# Can LLMs Correct Themselves? A Benchmark of Self-Correction in LLMs

**Guiyao Tie**[1]\*, **Zenghui Yuan**[1]\*, **Zeli Zhao**[1]\*, **Chaoran Hu**[1], **Tianhe Gu**[1],
**Ruihang Zhang**[1], **Sizhe Zhang**[1], **Junran Wu**[1], **Xiaoyue Tu**[1], **Ming Jin**[2],
**Qingsong Wen**[3], **Lixing Chen**[4]†, **Pan Zhou**[1]†, **Lichao Sun**[5]
[1]Huazhong University of Science and Technology    [2]Griffith University
[3]Squirrel Ai Learning    [4]Shanghai Jiaotong University    [5]Lehigh University
{tgy,zenghuiyuan,zhaozeli,huchaoran,gtianhe,ruihang_zhang}@hust.edu.cn
{zhangsizhe,junranw,xiaoyuetu,panzhou}@hust.edu.cn,
{mingjinedu,qingsongedu}@gmail.com,lxchen@sjtu.edu.cn,lis221@lehigh.edu

## Abstract

Self-correction of large language models (LLMs) emerges as a critical component for enhancing their reasoning performance. Although various self-correction methods have been proposed, a comprehensive evaluation of these methods remains largely unexplored, and the question of whether LLMs can truly correct themselves is a matter of significant interest and concern. In this study, we introduce **CorrectBench**, a benchmark developed to evaluate the effectiveness of self-correction strategies, including intrinsic, external, and fine-tuned approaches, across three tasks: commonsense reasoning, mathematical reasoning, and code generation. Our findings reveal that: 1) Self-correction methods can improve accuracy, especially for complex reasoning tasks; 2) Mixing different self-correction strategies yields further improvements, though it reduces efficiency; 3) Reasoning LLMs have limited optimization under additional self-correction methods and have high time costs. Interestingly, a comparatively simple chain-of-thought (CoT) baseline demonstrates competitive accuracy and efficiency. These results underscore the potential of self-correction to enhance LLM's reasoning performance while highlighting the ongoing challenge of improving their efficiency. Consequently, we advocate for further research focused on optimizing the balance between reasoning capabilities and operational efficiency. Project Page: https://correctbench.github.io/

## 1 Introduction

The rapid advancement of large language models (LLMs), exemplified by GPT-3.5 [72] and LLaMA 3 [16], has precipitated a transformative shift in artificial intelligence (AI), yielding state-of-the-art performance across diverse tasks [56]. Specifically, these tasks include content generation [1], natural language understanding [31], and complex decision-making [67], all of which have been revolutionized by the extensive pretraining and sophisticated architectures of LLMs. Notably, the introduction of frameworks like Chain-of-Thought (CoT) [61] has further expanded LLM's capacity for multi-step reasoning, enabling them to tackle more intricate tasks.

Despite these advancements, ensuring the reliability and accuracy of model outputs, especially for reasoning-intensive tasks, remains a formidable challenge. In response, recent works have focused on self-correction strategies aimed at refining LLMs' decision-making processes [29, 34] through iterative revision. Pioneering approaches such as RARR [20], Refiner [44], and CRITIC [21] illustrate the potential of integrating feedback loops and corrective components into model architectures.

---

\*Equal contributions.    †Corresponding authors: Pan Zhou and Lixing Chen.

39th Conference on Neural Information Processing Systems (NeurIPS 2025) Track on Datasets and Benchmarks.

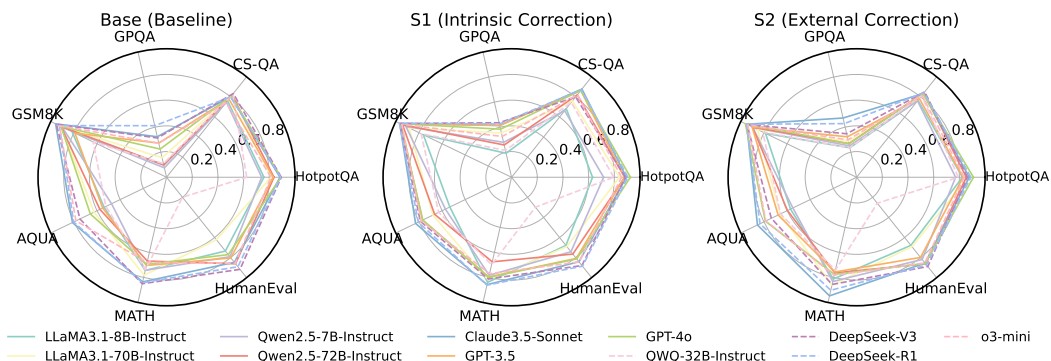

Figure 1: Comparison of different LLMs across various self-correction types and task scenarios.

However, these approaches often yield inconsistent gains across different tasks, prompting deeper questions about their capability of correction and generalizability. This observation motivates the central question: ***Can LLMs truly correct themselves?*** Moreover, it remains unclear whether more intricate self-correction schemes necessarily translate into superior overall performance.

To address these issues, this paper presents **CorrectBench**, a systematic benchmark for investigating how self-correction methods affect LLMs performance across multiple tasks. Building on a recent survey of self-correction approaches [29], we categorize such methods into three primary categories (i.e., *intrinsic correction*, *external correction* and *fine-tuned correction*), and select 11 representative methods from them. Additionally, we establish comparative baselines using both the widely adopted CoT [61] and a standard prompting strategy denoted as 'Base' (detailed in Appendix G.1).

For a rigorous and comprehensive assessment, we construct two curated datasets: *CorrectBench-base*, which integrates diverse subtasks with 3,825 question-answer pairs drawn from 7 distinct subdatasets, and *CorrectBench-test*, a curated collection of question-answer pairs specifically tailored for correction-oriented experiments. These subdatasets cover three principal tasks: commonsense reasoning [47], mathematical reasoning [23], and code generation [7]. We then apply these subdatasets to both instruction-based LLMs (e.g., LLaMA 3.1-8B-Instruct [40], Qwen 2.5-7B-Instruct [70], GPT-4o [26], Claude 3.5-Sonnet [2]) and reasoning LLMs[2] (e.g., DeepSeek-V3 [12]). Figure 1 compares the baseline ('Base') performance with the mean performances of intrinsic and external correction methods separately, revealing that self-correction bolsters overall accuracy (detailed in Figure 8).

**Key insights.** First, self-correction methods substantially enhance accuracy, particularly in complex reasoning tasks. Meanwhile, mixing multiple methods, while improving accuracy, incurs higher computational costs and reduced efficiency. For reasoning LLMs, these methods offer only marginal gains with increased time usage. Interestingly, the CoT [61] strategy demonstrates a favorable trade-off between operational efficiency and overall accuracy, challenging the prevailing assumption that more sophisticated correction frameworks inherently produce superior outcomes.

To summarize, our work provides three key contributions:

- **A Comprehensive Benchmark.** We propose CorrectBench, the first benchmark devised to systematically evaluate the impact of self-correction on LLMs inference. Spanning multiple tasks and model categories, CorrectBench offers a robust, reproducible platform for methodological comparisons.

- **Two Datasets.** We present CorrectBench-base and CorrectBench-test, both meticulously constructed to encompass a broad range of question-answer formats and reasoning scenarios, facilitating thorough assessments of different correction methods.

- **Insights and Implications.** Our empirical findings show that self-correction substantially advances LLMs' performance, especially on tasks demanding extensive reasoning. However, the increased computational load of mixing multiple correction strategies must be weighed against potential accuracy gains. Moreover, for reasoning LLMs, additional correction

---

[2]This paper defines "reasoning LLMs" as those models that are specifically enhanced with complex reasoning capabilities through a post-training optimization process.

methods provide limited improvements, emphasizing critical cost-efficiency concerns for practical applications.

# 2    CorrectBench: A Benchmark of Self-Correction in LLMs

CorrectBench is a systematically designed benchmark that quantifies the extent to which various correction methods improve model outputs in reasoning-intensive scenarios. As illustrated in Figure 2, CorrectBench characterizes self-correction along three principal dimensions: *Task Scenario*, *Self-Correction Type*, and *LLM Type*. The evaluation pipeline begins with selecting a specific task scenario and dataset, followed by applying a chosen correction method, and concludes with assessing the model's iterative self-correction process across diverse LLMs.

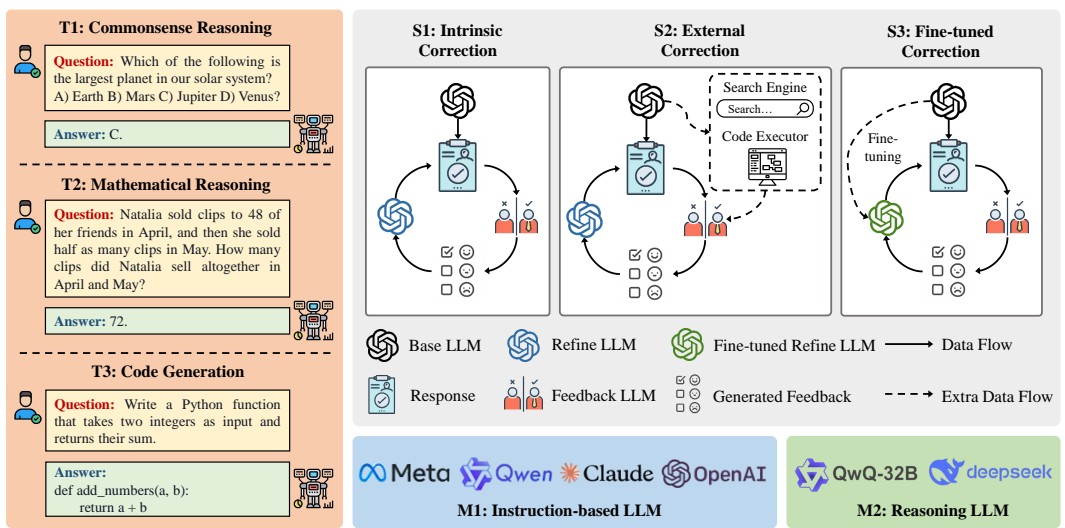

Figure 2: An overview of the CorrectBench framework.

**Iterative Self-Correction Paradigm.** In a standard LLM evaluation, the model generates an initial response $r_0$ to a question $q$ given an initial prompt $p_0$, formally $r_0 = \mathcal{M}(q, p_0)$, where $\mathcal{M}$ denotes the LLM. While this process becomes iterative in the self-correction paradigm. Specifically, for the $k$-th iteration, $p_k = p_{k-1} \cup r_{k-1}, r_k = \mathcal{M}(q, p_k)$, where $p_k$ is the updated prompt that includes the previous response $r_{k-1}$. After $K$ iterations, the final output $r_K$ reflects the model's *corrected* response. This iterative mechanism enables the model to continually refine its output based on newly revealed errors or inconsistencies.

**Mixture Framework.** While individual self-correction methods can improve model responses, it is plausible that integrating multiple methods may yield further improvements in accuracy. To examine these potential synergies, we propose the mixture framework, illustrated in Figure 3. The response of one correction method serves as input to the next, forming a dynamic pipeline of iterative refinements. This setup enables us to analyze how distinct self-correction methods in-

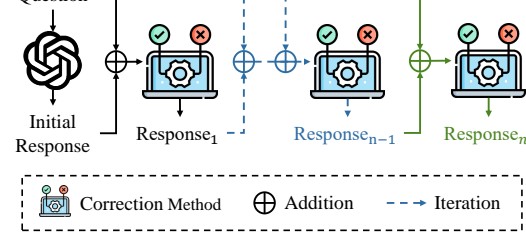

Figure 3: Mixture of different correction methods.

teract, thereby guiding the development of optimal configurations for improving LLM's performance.

## 2.1    Self-Correction Method

CorrectBench comprehensively evaluates three distinct categories of self-correction methods:

**S1: Intrinsic Correction.** This category focuses on the LLMs' capacity to internally identify and correct errors without external tools. Methods such as RCI [30], Self-Refine [38], CoVe [13], and Reflexion [54] enable the LLMs to re-evaluate its prior reasoning steps and resolve inconsistencies based on its internal knowledge.

**S2: External Correction.** In contrast to *S1*, *S2* (e.g., Reflexion [54], RARR [20], RATT [74], CRITIC [21]) leverages external resources, such as knowledge bases or Google search tools, to address gaps in the internal representation. This external support can correct factual inaccuracies or logical oversights, though it may constrain the model's capacity for divergent reasoning.

**S3: Fine-tuned Correction.** Fine-tuned correction methods including DCoT [45], SCORE [76], and SuperCorrect [69] enhance LLMs' self-correction performance through targeted fine-tuning. Although potentially effective, these methods require substantial training and are often limited by the scope and quality of the fine-tuning data.

## 2.2 Task Scenario

CorrectBench investigates self-correction methods across three representative task scenarios:

**T1: Commonsense Reasoning.** This scenario probes the model's capacity to address factual or logical inconsistencies within everyday knowledge domains. Relevant datasets include HotpotQA [71], CommonsenseQA [47][3], and the more challenging GPQA[50], which emphasizes complex reasoning.

**T2: Mathematical Reasoning.** Datasets in this task scenario emphasize the detection and correction of errors in mathematical derivations, algebraic manipulations, and multi-step reasoning. Representative datasets include GSM8K [11], AQUA [8], and MATH [23].

**T3: Code Generation.** This scenario assesses the LLM's ability to generate functionally correct and logically consistent code from natural language prompts. For instance, HumanEval [7] measures the LLM's ability to detect and correct syntax errors, logical flaws, and other coding mistakes.

## 2.3 LLM Type

To ensure a broad and realistic appraisal, CorrectBench encompasses two categories of LLMs:

**M1: Instruction-Based LLMs.** LLMs are designed to follow user-provided instructions to generate relevant outputs, making them highly versatile across diverse tasks. This category includes both open-source and closed-source LLMs, distinguished by their accessibility and design paradigms. Open-source LLMs, such as Qwen 2.5-7B/70B-Instruct [70], as well as LLaMA 3.1-8B/70B-Instruct [40], offer transparency and flexibility for modification, enabling detailed analysis and fine-tuning. Conversely, closed-source LLMs, including OpenAI's GPT-3.5 [42], GPT-4o [26], and Anthropic's Claude 3.5-Sonnet [2], excel in real-world tasks due to proprietary optimizations, but restrict direct access and customization for research purposes.

**M2: Reasoning LLMs.** Reasoning LLMs are models specifically enhanced with advanced reasoning capabilities through targeted post-training optimization processes. These models are designed to excel in tasks requiring multi-step logical reasoning, often incorporating integrated self-correction mechanisms to refine their outputs. Representative examples include QWQ-32B-Instruct [66], o3-mini and DeepSeek-R1 [12]. DeepSeek-V3 [12], in particular, adopts an innovative approach to distilling reasoning capabilities from long chain-of-thought models, leveraging its predecessor. By integrating verification and reflection patterns from R1, DeepSeek-V3 achieves substantial improvements in reasoning accuracy while maintaining precise control over output style and length.

## 2.4 Research Question

This study aims to elucidate the effectiveness of different self-correction methods in enhancing LLMs' performance, addressing the following core research questions:

**[RQ1]** To what extent can LLMs achieve accurate results by leveraging intrinsic (*S1*) and external (*S2*) self-correction methods[4] without requiring further intervention?

**[RQ2]** How does mixing multiple self-correction methods influence model accuracy and robustness, and what are the associated computational trade-offs?

**[RQ3]** For reasoning LLMs with built-in correction mechanisms, to what extent can the above self-correction methods provide additional benefits?

---

[3]CommonsenseQA is represented as CS-QA in the following.

[4]*S3* is analyzed separately due to dataset-specific constraints.

Table 1: Main results on CorrectBench for the average of multiple LLMs. Values in () indicate the change from the baseline. Blue signifies improvements, and orange indicates declines, where darker shades reflect larger magnitudes. Further details are given in Appendix H.

| Type | Method | HotpotQA(↑) | CS-QA(↑) | GPQA(↑) | GSM8K(↑) | AQUA(↑) | MATH(↑) | HumanEval(↑) |
|---|---|---|---|---|---|---|---|---|
| - | Base | 80.76 | 79.96 | 18.56 | 86.46 | 61.23 | 75.12 | 72.71 |
| | CoT | 83.29 ( +2.53 ) | 78.03 ( -1.93 ) | 16.52 ( -2.04 ) | 91.96 ( +5.50 ) | 60.24 ( -0.99 ) | 72.59 ( -2.53 ) | 60.10 ( -12.61 ) |
| S1 | RCI | 79.67 ( -1.09 ) | 76.29 ( -3.67 ) | 19.98 ( +1.42 ) | 87.00 ( +0.54 ) | 67.12 ( +5.89 ) | 74.92 ( -0.20 ) | 67.46 ( -5.25 ) |
| | CoVe | 83.04 ( +2.28 ) | 78.54 ( -1.42 ) | 37.41 ( +18.85 ) | 92.23 ( +5.77 ) | 71.12 ( +9.89 ) | 79.30 ( +4.18 ) | 76.96 ( +4.25 ) |
| | Self-Refine | 85.49 ( +4.73 ) | 81.06 ( +1.10 ) | 40.69 ( +22.13 ) | 91.74 ( +5.28 ) | 69.46 ( +8.23 ) | 81.77 ( +6.65 ) | - |
| | Reflexion-v1 | 69.52 ( -11.24 ) | 63.89 ( -16.07 ) | 19.25 ( +0.69 ) | 67.64 ( -18.82 ) | 48.33 ( -12.90 ) | 65.01 ( -10.11 ) | - |
| S2 | Reflexion-v2 | 87.98 ( +7.22 ) | 82.21 ( +2.25 ) | 26.85 ( +8.29 ) | 89.87 ( +3.41 ) | 68.23 ( +7.00 ) | 81.36 ( +6.24 ) | - |
| | RARR | 85.47 ( +4.71 ) | 80.57 ( +0.61 ) | 36.82 ( +18.26 ) | 88.92 ( +2.46 ) | 66.81 ( +5.58 ) | 82.78 ( +7.66 ) | 77.35 ( +4.64 ) |
| | RATT | 79.59 ( -1.17 ) | 80.81 ( +0.85 ) | 25.90 ( +7.34 ) | 88.08 ( +1.62 ) | 68.06 ( +6.83 ) | 80.74 ( +5.62 ) | 73.44 ( +0.73 ) |
| | CRITIC | - | 81.77 ( +1.81 ) | - | 77.46 ( -9.00 ) | - | - | - |
| - | Average | 83.54 ( +2.78 ) | 80.18 ( +0.22 ) | 31.28 ( +12.72 ) | 85.04 ( -1.42 ) | 68.47 ( +7.24 ) | 80.15 ( +5.03 ) | 73.80 ( +1.09 ) |

# 3 Experiment Settings

**Dataset Preparation.** To ensure consistency and reproducibility, we employ *CorrectBench-test* for experimental evaluations. For each dataset within CorrectBench-test, we randomly select 100 samples and subsequently refine this selection by excluding a minimal number of outliers or irrelevant instances, thereby ensuring a more precise representation of error patterns. Comprehensive details regarding the datasets and preprocessing steps are provided in Appendix B.1, thereby promoting transparency and reproducibility for subsequent research.

**Task and Model Selection.** CorrectBench adopts a hierarchical strategy to evaluate self-correction across diverse task scenarios and LLM types. Specific datasets are selected to match the characteristics and objectives of each self-correction method, ensuring that the benchmark captures comprehensive error types and correction challenges. Further specifications regarding selections of datasets and LLMs are included in Appendix B.2, ensuring full reproducibility of the evaluation methodology.

**Evaluation Metrics.** We employ both task-specific and judgment-based metrics to evaluate the self-correction capabilities of diverse tasks: **1) Task-Specific Metrics.** These metrics are tailored to evaluate model performance across different tasks. For *T1*, accuracy is computed as: $\text{ACC} = \frac{1}{N} \sum_{i=1}^{N} \mathbb{I}(\hat{y}_i = y_i)$, where $N$ is the total number of samples, $y_i$ is the ground truth, $\hat{y}_i$ is the predicted answer, and $\mathbb{I}(\cdot)$ is the indicator function. For *T2*, the solve rate represents the percentage of problems correctly solved by the model out of the total number of problems. For *T3*, pass@k evaluates whether at least one of the $k$ generated solutions for a problem passes all test cases. The final pass@k score is calculated as the average pass rate across all problems. **2) Judgment-Based Metrics.** In cases where the model's response is ambiguous or incomplete, we conduct human evaluations, where human evaluators apply stringent criteria to ensure an impartial and thorough assessment of the judgments. Additionally, we employ GPT-4o as an *LLM-as-a-Judge* paradigm, providing an automated yet consistent scoring mechanism for large-scale experimental runs.

# 4 Empirical Results and Analysis

## 4.1 Main Results

Table 1 summarizes the average performance improvements attained by various self-correction methods over the 'Base'. The results reveal that each self-correction method demonstrates performance improvements over the 'Base' to varying degrees, with particularly pronounced gains in more complex tasks such as GPQA and MATH. For instance, CoVe from *S1* yields an improvement of +23.24% on GPQA. However, simpler tasks like GSM8K exhibit more modest gains (e.g., +5.28% for CoVe). By contrast, external correction methods *S2* generally achieve higher average gains than *S1*. For example, Reflexion-v1[5] experiences declines on tasks such as HotpotQA (-11.13%) and AQUA (-12.90%). However, Reflexion-v2[6] increases its effectiveness, yielding improvements of +7.33% on HotpotQA and +7.00% on AQUA. We analyze that is because Reflexion [54] was initially

---

[5]Reflexion-v1 denotes reflexion without external tools

[6]Reflexion-v2 denotes reflexion with external tools

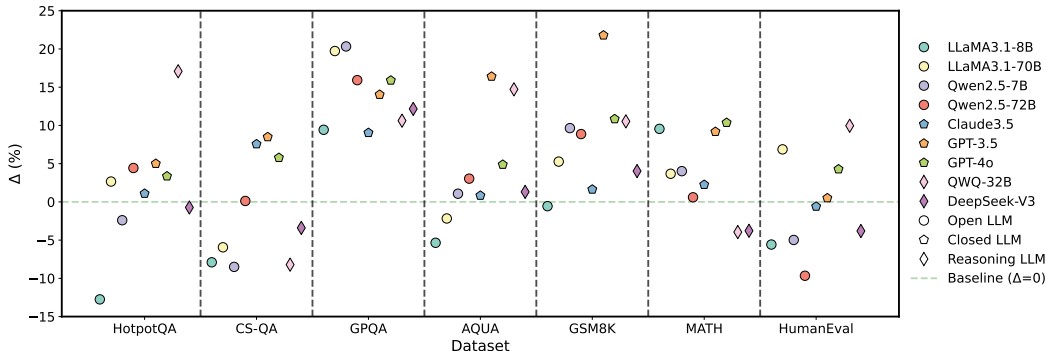

Figure 4: Average performance improvements achieved by *S1* across multiple LLMs.

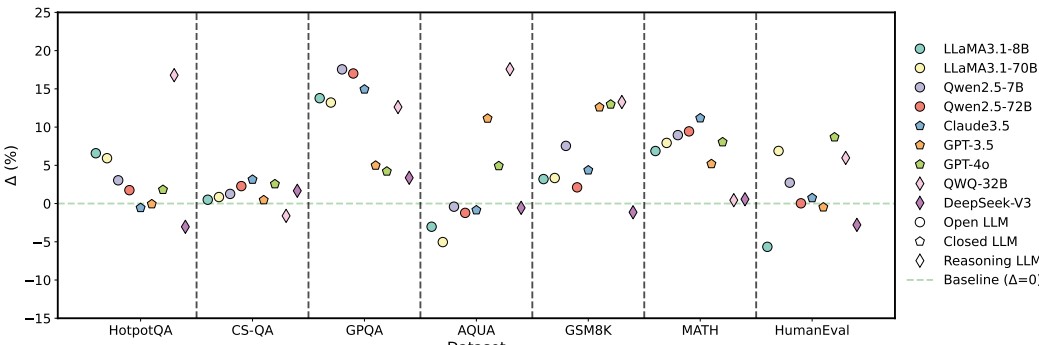

Figure 5: Average performance improvements achieved by *S2* across multiple LLMs.

designed to leverage external tools for enhanced correction, but Reflexion-v1, stripped of these tools, lacks optimization. This leaves weaker LLMs prone to getting "stuck," producing persistent incorrect outputs and lowering the average score. Finally, *S3* generally lags behind these methods, likely due to smaller model sizes and narrower training objectives (Details in Section 4.5).

## 4.2 Results of Intrinsic Correction

Figure 4 illustrates the mean performance gains realized by all *S1* methods across nine LLMs and multiple datasets, where *y=0* denotes the baseline. For detailed results of each method, refer to Appendix E.1. Although *S1* improves accuracy overall, the degree of improvement varies across instruction-based and reasoning LLMs. **1) Instruction-based LLMs.** Closed-source LLMs exhibit uneven performance gains. For instance, LLaMA3.1-8B-Instruct shows significant declines on AQUA and HumanEval, whereas Qwen2.5-7B-Instruct demonstrates modest gains. These discrepancies likely stem from smaller parameter sizes and weaker instruction-following capabilities. In contrast, open-source LLMs offer more consistent and stable performance improvements. Notably, GPT-4o shows substantial gains on GPQA, whereas Claude3.5 achieves similar enhancements, highlighting the robust instruction-following adaptability of open-source architectures. **2) Reasoning LLMs.** DeepSeek-V3's performance remains close to the baseline across most tasks. While it demonstrates slight improvements on datasets such as CS-QA, GPQA, and MATH, it exhibits marginal declines on others. To further investigate this phenomenon, we conducted additional experiments (see Section 4.8), revealing that DeepSeek-V3's built-in correction mechanism delivers a strong baseline performance. This high initial performance likely limits the impact of other correction methods. Conversely, QWQ attains considerable improvements on most datasets except CS-QA and MATH, likely reflecting the constraints posed by its smaller parameter size.

## 4.3 Results of External Correction

Figure 5 illustrates the average performance improvements achieved by each LLM utilizing external correction methods (*S2*) across various datasets. The trends observed align closely with those depicted in Figure 4, indicating a consistent enhancement in overall performance. Notably, external correction methods demonstrate greater stability, which can be attributed to their reliance on authoritative

external resources. By referencing these resources, *S2* effectively mitigates the occurrence of incorrect responses. However, this reliance on external inputs may also limit the LLM's capacity for divergent or creative reasoning, resulting in steadier but less flexible performance compared to intrinsic correction methods. For comprehensive results for each method, please refer to Appendix E.2.

> **[RQ1]** To what extent can LLMs achieve accurate results by leveraging intrinsic (*S1*) and external (*S2*) self-correction methods without requiring further intervention?
> **Conclusion:** Both *S1* and *S2* enable significant performance gains, particularly for complex tasks requiring multi-step reasoning or domain-specific knowledge. By iteratively refining responses, these methods effectively correct themselves even without additional fine-tuning.

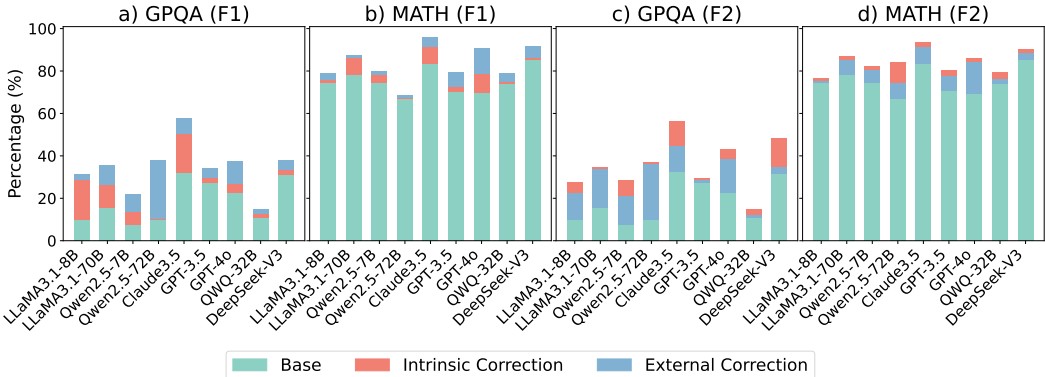

Figure 6: Comparison of different LLMs with mixture methods.

## 4.4 Results of Mixture Method

We further explored whether mixing multiple self-correction methods results in additive or synergistic performance improvements. Specifically, the responses generated by the baseline method (*Base*) are utilized as auxiliary prompts for an intrinsic method (*S1*), an external method (*S2*), or a mixture of both (*S1+S2*). As shown in Figure 6, we evaluated two representative configurations (e.g., F1: 'Base to *S1* to *S2*' and F2: 'Base to *S2* to *S1*') on two benchmark tasks: GPQA and MATH. The results revealed two key findings. Firstly, applying one or more correction methods consistently improves model performance to varying degrees. Secondly, *S2* generally yields larger performance gains compared to *S1*. Notably, complex reasoning tasks, such as GPQA, benefit the most from these mixed methods. However, these mixtures often introduce additional computational overhead. To address the associated efficiency and accuracy trade-offs, we conducted a complementary analysis of response times under different correction methods and model configurations, as detailed in Section 4.7.

## 4.5 Results of Fine-tuned Correction

Table 2 summarizes the performance of fine-tuned correction methods (*S3*), revealing two main observations. First, *S3* (e.g., DCoT) often exhibits inconsistent outcomes across diverse tasks. This variability stems from their reliance on fine-tuning with narrowly focused datasets, which restricts their broader applicability. Second, domain-specific fine-tuning proves especially promising for models tailored to specialized tasks. For example, SuperCorrect, fine-tuned on Qwen2.5-Math-7B-Instruct, demonstrates marked improvements in mathematical reasoning (e.g., on GSM8K and MATH), outperforming other methods by a wide margin. This underscores the effectiveness of leveraging task-aligned models, particularly when fine-tuning objectives closely align with the target domain requirements.

Table 2: Performance of fine-tuned methods on selected datasets.

| Fine-tuned LLM | Method | CS-QA(↑) | GSM8K(↑) |
|---|---|---|---|
| - | Base | 31.40 | 56.75 |
| LLaMA2-7B-hf | DCoT | 29.65(-1.75) | 41.20(-15.55) |
| Gemma-7B-it | SCORE | 43.26(+11.86) | 75.30(+18.55) |
| LLaMA2-13B-chat | SCORE | 41.45(+10.05) | 72.10(+15.35) |
| Qwen2.5-Math-7B-Instruct | SuperCorrect | 46.25(+14.85) | 84.30(+27.55) |
| | | **MATH(↑)** | **HumanEval(↑)** |
| - | Base | 41.71 | 26.25 |
| Qwen2.5-Math-7B-Instruct | SuperCorrect | 70.16(+28.45) | 39.30(+13.05) |

## 4.6 Results of Correction and Misjudgment

In order to further assess the correction ability, we divide the responses of different models to questions in the three most challenging tasks (GPQA, AQUA, and HotpotQA) into error-based dataset and correction-based dataset, corresponding to the wrong and correct question-response pairs, respectively. We select CoVe and RARR from *S1* and *S2* respectively to evaluate on Claude 3.5-Sonnet. We hereby define two new metrics: **Correction Rate (CR)** indicates the proportion of incorrect responses that are successfully corrected, and **Misjudgment Rate (MR)** refers to the proportion of correct responses that are misjudged to be corrected wrongly. The results shown in Table 3 reflects that both methods achieve high CRs and low MRs, which shows that self-correction methods can effectively correct the wrong examples with less misjudgment.

Table 3: Performance of correction rate and misjudgment rate.

| Method | Rate | GPQA | AQUA | HotpotQA | Overall |
|--------|------|------|------|----------|---------|
| Cove   | CR   | 31.6 | 36.0 | 52.1     | 40.8    |
|        | MR   | 8.1  | 8.0  | 6.7      | 7.5     |
| RARR   | CR   | 30.7 | 49.3 | 51.3     | 47.1    |
|        | MR   | 5.5  | 4.3  | 4.5      | 4.5     |

## 4.7 Results of Response Time

Figure 7 compares the average response times across representative models (e.g., LLaMA3.1-70B and GPT-4o from *M1*, DeepSeek-V3 from *M2*) under various correction methods. In general, both intrinsic and external methods extend inference times relative to baseline approaches (Base and CoT), largely due to increased reasoning complexity or reliance on external services (especially for RATT). Moreover, reasoning LLMs, such as DeepSeek-V3, exhibit notably longer execution times than instruction-based models, likely attributable to their built-in correction mechanisms. In contrast, the baseline CoT method achieves notably shorter response times while maintaining a reasonable accuracy (combined with Table 1). This observation underscores that more complex correction strategies

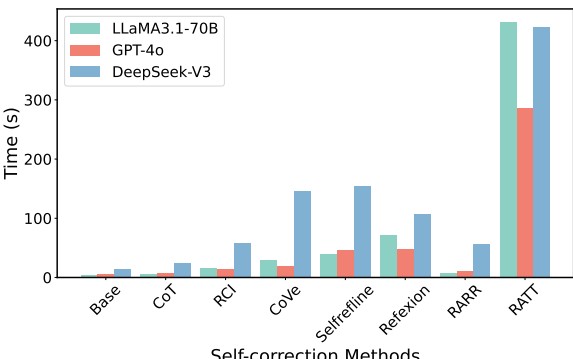

Figure 7: Average response times for LLaMA3.1-70B, GPT-4o, and DeepSeek-V3. Intrinsic (*S1*) and external (*S2*) methods generally increase inference duration relative to Base and CoT.

do not always yield superior outcomes, highlighting the critical trade-off between model accuracy and computational efficiency. As shown in Table 4, RARR offers a balanced trade-off between efficiency and accuracy, with only 533 tokens and 2 API calls. Reflexion-v2 and RATT achieve the highest accuracies, reflecting the benefit of external retrieval or code execution, while their overhead remains manageable (below 15% additional search tokens). Bootstrap-based confidence intervals for these results are reported in Appendix B.3.

> **[RQ2]** How does mixing multiple self-correction methods influence model accuracy and robustness, and what are the associated computational trade-offs?
> **Conclusion:** Mixing self-correction methods typically results in accuracy improvements, though at the cost of increased computational overhead. Such mixtures are particularly beneficial for high-precision tasks where the trade-off of more runtime is justifiable.

## 4.8 Results of Reasoning LLMs

To further investigate why DeepSeek-V3 shows limited improvement from self-correction, we compare instruction-based LLMs and reasoning LLMs on the *Base* alone. Table 5 shows that DeepSeek-V3 consistently achieves top-2 or even top-1 performance across all datasets. Combined with the results of Section 4.2, we find that this may be because reasoning LLMs already incorporate robust intrinsic correction mechanisms, limiting additional gains from external correction steps. In particular, DeepSeek-V3 integrates advanced reflection modules and comprehensive error-detection

Table 4: Comprehensive resource cost analysis on the MATH dataset (150 samples). Values in **green** indicate the best trade-off between cost and accuracy, while **red** highlights the least efficient results. "Efficiency Rank" represents the ratio of Accuracy / (Token Count × API Calls).

| Method | Type | Avg. Tokens | API Calls | Search Tokens (%) | Accuracy (%) | Efficiency Rank |
|--------|------|-------------|-----------|-------------------|--------------|-----------------|
| Base | - | 791 | 1.0 | 0 (0%) | 68.5 | 0.0866 |
| CoT | - | 1804 | 1.0 | 0 (0%) | 69.5 | 0.0385 |
| CoVe | S1 | 2019 | 1.0 | 0 (0%) | 75.0 | 0.0371 |
| RCI | S1 | 1780 | 1.2 | 0 (0%) | 70.2 | 0.0328 |
| Reflexion-v1 | S1 | 1460 | 3.5 | 0 (0%) | 72.8 | 0.0143 |
| Reflexion-v2 | S2 | 1712 | 4.0 | 154 (8.25%) | 74.5 | 0.0109 |
| RARR | S2 | **533** | 2.0 | 89 (14.31%) | 76.3 | **0.0716** |
| RATT | S2 | 2185 | 3.0 | 162 (6.9%) | **78.7** | 0.0120 |

Table 5: Comparison of baseline performance among instruction-based and reasoning LLMs. Per-column maxima are **bolded**; per-column minima are underlined. Per-row maxima are highlighted with blue; per-row minima are highlighted with orange.

| Type | LLM | HotpotQA | CS-QA | GPQA | GSM8K | AQUA | MATH | HumanEval |
|------|-----|----------|-------|------|-------|------|------|-----------|
| **Open-source** | LLaMA3.1-8B-Instruct | 75.80 | 76.16 | 9.74 | 81.55 | 53.88 | 74.37 | 73.44 |
| | LLaMA3.1-70B-Instruct | 81.28 | 81.88 | 15.62 | 90.63 | 62.65 | 78.21 | 62.18 |
| | Qwen2.5-7B-Instruct | 74.05 | 74.75 | 7.53 | 90.23 | 47.50 | 74.28 | 79.11 |
| | Qwen2.5-72B-Instruct | 83.63 | 81.92 | 9.85 | 91.11 | 57.58 | 66.91 | 86.13 |
| **Closed-source** | Claude3.5-Sonnet | 88.29 | 80.25 | 32.34 | **95.81** | **81.26** | 83.51 | 84.69 |
| | GPT-3.5 | 82.94 | 77.92 | 27.29 | 79.14 | 55.15 | 70.44 | 80.29 |
| | GPT-4o | 89.16 | 80.65 | 22.49 | 91.15 | 65.82 | 69.54 | 77.04 |
| **Reasoning** | QWQ-32B | 62.43 | 82.78 | 10.85 | 63.41 | 52.42 | 73.78 | 19.86 |
| | DeepSeek-V3 | **89.29** | **83.35** | 31.35 | 95.12 | 74.79 | **85.02** | **91.67** |
| | DeepSeek-R1 | 88.92 | 79.93 | **41.15** | 92.63 | 80.23 | 84.21 | 89.06 |
| | o3-mini | 81.24 | 74.28 | 27.17 | 92.45 | 78.26 | 67.97 | 85.75 |

routines distilled from its earlier R1 series, enabling thorough multi-step reasoning at the outset. This high baseline effectively reduces the scope for further improvement through additional self-correction. Consequently, attempts to augment DeepSeek-V3 with further self-correction methods produce minimal net gains while incurring additional computational overhead.

> **[RQ3]** For reasoning LLMs with built-in correction mechanisms, to what extent can the above self-correction methods provide additional benefits?
> **Conclusion:** Reasoning LLMs (e.g., DeepSeek-V3) already embed sophisticated error-detection and correction processes. As a result, additional self-correction methods confer only marginal gains and may increase computational overhead, highlighting a performance ceiling in highly reasoning LLMs.

### 4.9 Failure Mode Taxonomy and Case Analysis

To better understand why different self-correction strategies succeed or fail, we conducted a supplemental failure-mode analysis on the GPQA (250 samples) and MATH (500 samples) datasets. Six major categories of failure were identified, alongside a residual "Other" category, as summarized in Table 6. Logical Oversight (32.9%) and Factual Inaccuracy (22.0%) dominate, implying that intrinsic corrections (S1) are suitable for reasoning-related errors, while external corrections (S2) excel at factual validation. These findings motivate our adaptive correction controller (Sec. 6), which dynamically selects correction strategies based on detected failure types.

## 5 Related Work

**Self-Correction Methods.** With the continuous development of self-correction techniques [62, 65, 18, 32, 63], researchers have proposed various approaches to enhance the performance of large language models. Intrinsic methods, such as CoVe [13] and RCI [30], improve the precision and

Table 6: Error taxonomy of LLM self-correction failures across GPQA and MATH. Logical and factual errors dominate, suggesting distinct correction strategies (S1 vs. S2).

| Failure Mode | Pro. (%) | Description | Suggested Correction |
|---|---|---|---|
| Logical Oversight | 32.9 | Reasoning step errors (e.g., misapplied formula) | S1: CoT, RCI |
| Factual Inaccuracy | 22.0 | Outdated or incorrect retrieved evidence | S2: RARR, RATT |
| Over-Reliance on Tools | 14.6 | Excessive external API calls causing inefficiency | S2 (bounded) |
| Ambiguous Output | 14.2 | Incomplete or vague final answer | S1 refinement |
| Contextual Misunderstanding | 10.8 | Misinterpreted question or missing context | S1+S2 hybrid |
| Computational Error | 3.5 | Faulty code execution or symbolic computation | S2 verification |
| Other | 2.0 | Miscellaneous or formatting issues | - |

consistency of generated content through self-supervised mechanisms within the model. At the same time, extrinsic methods, such as CRITIC [21], RATT [74], and RARR [20], rely on an external tool to evaluate and provide feedback on the generated outputs, guiding the model towards optimization. Fine-tuned methods, such as DCoT [45], Supercorrect [69], and SCORE [76], further enhance the performance of the model by fine-tuning it for specific tasks, enabling more accurate and efficient handling of complex tasks. The continuous evolution of these methods provides diverse options and techniques for self-correction. Detailed discussions on the related word are provided in Appendix C.

**Correction Benchmarks.** Benchmarking the LLMs' self-correction ability [57, 15, 77, 14, 5, 79, 53, 37] has prompted the development of specialized benchmarks for different tasks. For instance, CriticBench [34] evaluates critique ability using discrimination results, but it struggles with task-specific fine-grained metrics and reliance on costly human annotations or potentially biased GPT-4 outputs. In the realm of vision-language models, VISCO [64] focuses on self-correction in multimodal tasks, while Beyond Correctness [78] specifically targets self-correction in large models for code generation. Our CorrectBench focused on striking a trade-off between reasoning capability and efficiency, proposing more generalized and nuanced evaluation methods for complex reasoning tasks.

# 6 Future Improvements in Self-Correction

Looking ahead, several promising directions can further enhance the robustness and adaptability of self-correction in large language models. (1) **Dynamic Adjustment:** reinforcement learning or meta-controller mechanisms could dynamically select among correction strategies (S1–S3) based on task complexity or confidence levels, reducing redundant computation through early stopping. (2) **Task-Specific Optimization:** domain-oriented fine-tuning, such as the 10.2% improvement achieved by SUPERCORRECT on MATH, suggests the value of adaptive pipelines that align with domain reasoning depth and structure. (3) **Human-in-the-Loop Integration:** in sensitive fields like medicine or law, coupling automated correction with limited expert feedback could improve factual reliability and ensure accountable model behavior. (4) **Meta-Controller Framework:** developing a lightweight controller to detect and correct intermediate reasoning inconsistencies may help refine the chain-of-thought process and prevent logical drift. Further analysis and discussions of these future directions are provided in Appendix D.

# 7 Conclusion

This paper presents **CorrectBench**, a comprehensive and extensible benchmark for evaluating the self-correction capabilities of large language models (LLMs) across diverse reasoning-intensive tasks, including commonsense inference, mathematical problem-solving, and code generation. Through systematic evaluation, we demonstrate that modern LLMs are increasingly capable of genuine self-correction, with reasoning-oriented models such as DEEPSEEK-R1 achieving substantial baseline accuracy and showing consistent improvement through iterative refinement. CorrectBench not only reveals the effectiveness of various correction paradigms (S1, S2, S3) but also exposes critical limitations—such as diminishing returns in deeper correction chains and resource inefficiencies in web-augmented methods. These insights emphasize the necessity for adaptive, cost-aware correction mechanisms that balance efficiency and reasoning depth. Overall, this study provides a unified foundation for understanding and benchmarking LLM self-correction. We hope this work serves as a stepping stone toward more trustworthy and self-improving language models.

## Acknowledgments

This work is supported by National Natural Science Foundation of China (NSFC) under grant No. 62476107.

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

# A Overview of Correction Performance

Figure 8 shows the comparison between the performance of the baseline and the average performance of the intrinsic correction and external correction methods on different tasks on different LLMs. It can be observed that both intrinsic correction and external correction outperform the baseline on most models and tasks.

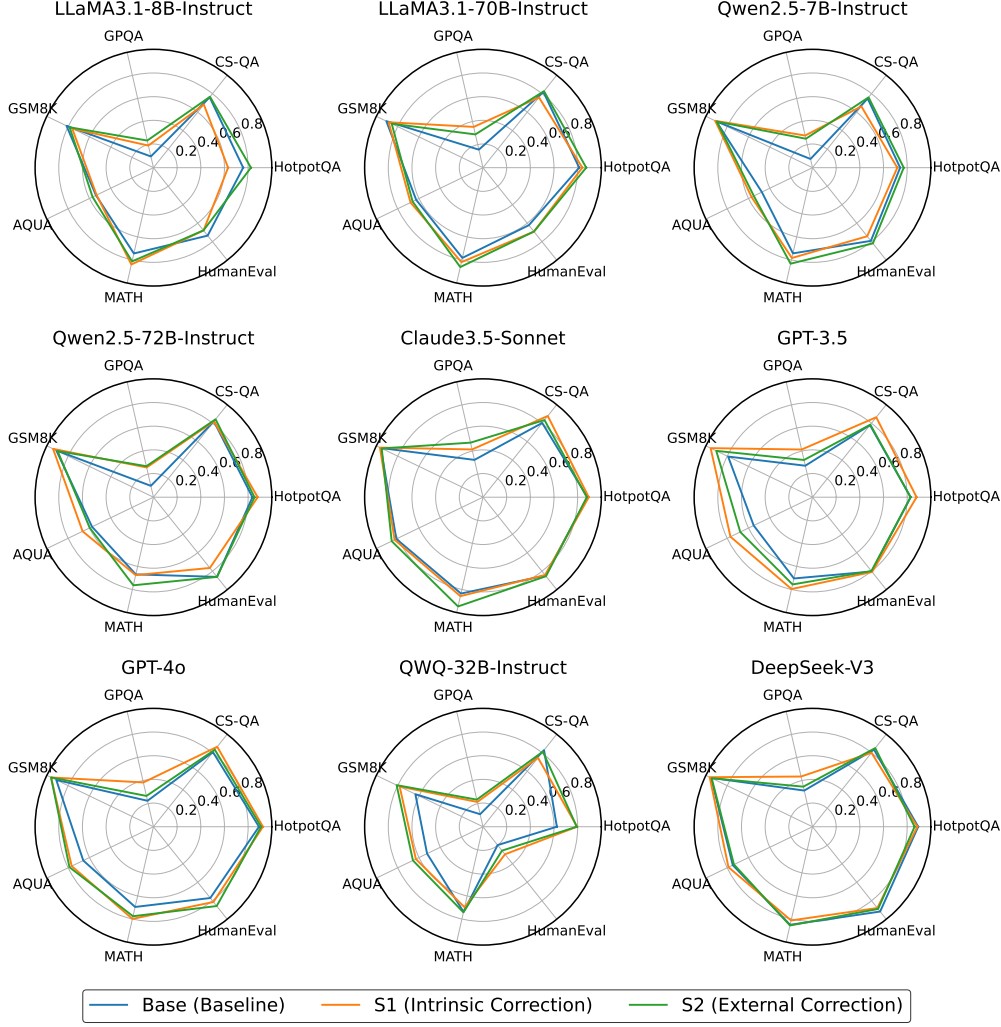

Figure 8: Comparative performance of different LLMs across various self-correction types and task scenarios.

# B Dataset

## B.1 Data Setting

This benchmark includes datasets from various domains as shown in Table 7: GSM8K, AQuA, and MATH for mathematical reasoning, HotpotQA, CommonsenseQA, and GPQA for commonsense reasoning, and HumanEval for code generation. GSM8K and AQuA feature high school-level math and quantitative reasoning problems, while MATH provides a broader set of mathematical challenges. HotpotQA and CommonsenseQA (CS-QA) test multi-hop and commonsense reasoning, respectively, with GPQA expanding on the latter by including more diverse questions. HumanEval consists of programming problems to assess code generation abilities.

Table 7: Statistics of the datasets used in CORRECTBENCH.

| Type | Dataset | Samples | License |
|------|---------|---------|---------|
| Commonsense | HotpotQA | 300 | CC BY-SA 4.0 |
| | CommonsenseQA | 300 | MIT License |
| | GPQA | 250 | Apache License 2.0 |
| Math | GSM8K | 250 | MIT License |
| | AQuA | 254 | Apache License 2.0 |
| | MATH | 500 | MIT License |
| Coding | HumanEval | 164 | MIT License |
| **All** | - | 2018 | - |

To ensure consistency and reproducibility, each dataset is sampled to include 100 examples, selected through a randomized process. To refine the dataset quality, we remove outliers or irrelevant samples, ensuring a more accurate representation of typical error patterns. For mathematical reasoning datasets such as GSM8K, AQuA, and MATH, we ensure that selected problems span diverse difficulty levels to capture a comprehensive assessment of model performance. Similarly, commonsense reasoning datasets (HotpotQA, CS-QA, and GPQA) are curated to include a balanced mix of multi-hop and diverse reasoning tasks. For HumanEval, programming problems are filtered to maintain relevance to standard coding scenarios while avoiding overly specialized or ambiguous cases.

## B.2 Dataset and LLM Selection

Table 8 summarizes the experimental evaluation of various self-correction methods across multiple datasets. The '✓' indicates that the corresponding method is evaluated on the dataset, whereas the '-' signifies that there are no experiments.

Table 8: Selection of different datasets and LLMs for all self-correction methods.

| Type | Methods | T1 | | | T2 | | | T3 | LLM Type | |
|------|---------|----------|-------|------|-------|------|------|-----------|----|----|
| | | HotpotQA | CS-QA | GPQA | GSM8K | AQUA | MATH | HumanEval | M1 | M2 |
| S1 | RCI | ✓ | ✓ | ✓ | ✓ | ✓ | ✓ | ✓ | ✓ | ✓ |
| | CoVe | ✓ | ✓ | ✓ | ✓ | ✓ | ✓ | ✓ | ✓ | ✓ |
| | Self-Refine | ✓ | ✓ | ✓ | ✓ | ✓ | ✓ | - | ✓ | ✓ |
| | Reflexion-v1 | ✓ | ✓ | ✓ | ✓ | ✓ | ✓ | - | ✓ | ✓ |
| S2 | Reflexion-v2 | ✓ | ✓ | ✓ | ✓ | ✓ | ✓ | - | ✓ | ✓ |
| | RARR | ✓ | ✓ | ✓ | ✓ | ✓ | ✓ | ✓ | ✓ | ✓ |
| | RATT | ✓ | ✓ | ✓ | ✓ | ✓ | ✓ | ✓ | ✓ | ✓ |
| | CRITIC | - | ✓ | - | ✓ | - | - | - | ✓ | ✓ |
| S3 | DCoT | - | ✓ | - | ✓ | - | - | - | - | - |
| | SCORE | - | ✓ | - | ✓ | - | - | - | - | - |
| | Supercorrect | - | ✓ | - | ✓ | - | ✓ | ✓ | - | - |

## B.3 Bootstrap Confidence Intervals for Resource Cost Results

To validate the robustness of the trade-offs in Table 4, we report 95% bootstrap confidence intervals for each method's accuracy across five random subsamples. The mean accuracy difference between S1 and S2 methods is statistically significant ($p < 0.05$), confirming that cost-normalized accuracy scales with correction depth.

# C Comprehensive Related Works

## C.1 Self-Correction Methods

**Theoretical Perspectives.** Recent research has delved into the theoretical foundations of self-correction in large language models (LLMs), particularly examining how these models refine their outputs through iterative self-examination. Key transformer components, such as softmax attention and multi-head attention, have been identified as central mechanisms in enabling this self-correction process [60]. Several studies have highlighted the limitations of intrinsic self-correction. For instance, LLMs often encounter challenges when attempting to self-correct reasoning tasks without external feedback, resulting in degraded performance in specific scenarios [25]. Nevertheless, other research has demonstrated that intrinsic self-correction can be effective under certain conditions, such as employing zero-temperature settings and fair prompts. These conditions help LLMs enhance accuracy across various tasks by providing a more structured and deterministic framework for response refinement [35]. Further investigations reveal that intrinsic self-correction processes can converge over multiple iterations, yielding stable performance improvements, particularly in iterative and complex tasks [36]. However, some studies challenge the notion that LLMs can consistently enhance their outputs through self-correction alone. These findings suggest that LLMs often struggle to differentiate between previously generated alternatives, which limits the effectiveness of their self-correction mechanisms [27]. Additionally, innovative approaches such as the Divergent CoT (DCoT) method have been proposed. By generating and comparing multiple divergent reasoning chains, this method improves reasoning accuracy and facilitates more effective self-correction during complex reasoning tasks [45].

**Self-Detection of Mistakes.** Self-detection of mistakes in LLM responses, often with the aid of external information, has been widely explored across several domains. In misinformation detection, numerous studies have investigated how LLMs can identify and correct errors in the information they generate [75, 10, 6, 41]. Similarly, context-faithfulness, which examines whether LLMs maintain consistency with the context in which they are deployed, has also been a focal point in recent research [59, 17, 52]. Other works have concentrated on harmful content detection, where LLMs are tasked with identifying potentially harmful or offensive outputs [49], as well as bias detection, which aims to identify and mitigate biases in LLM responses [4, 19]. Despite significant progress, recent studies have shown that even state-of-the-art LLMs struggle to reliably detect their own mistakes across a variety of tasks [58, 28]. For instance, research demonstrates that LLMs often fail to identify errors in their outputs, even when performing complex reasoning or content generation tasks. These findings highlight a crucial gap in the current self-correction capabilities of LLMs, underscoring the need for further research into more robust error detection and correction mechanisms.

**Fine-tuning Methods.** Self-training, or self-improvement, involves models utilizing their own responses to enhance performance. Several studies have explored the use of self-evaluation or self-correction for generating training data. For example, [3] and [22] leverage self-correction as a means to create training datasets, while [43] employ self-evaluation as a training signal to improve model performance. Another direction within self-training focuses on improving reasoning in LLMs by selecting high-quality generated outputs. [73] enhance reasoning by selecting outputs based on ground-truth final answers, whereas [24] emphasize self-consistency as a method for refining reasoning. [39] adopt a different approach by using high-confidence sentences generated by LLMs to train classifiers, demonstrating the potential of leveraging model confidence in improving task performance.

## C.2 Correction Benchmarks.

Benchmarking the LLMs' self-correction ability has prompted the development of specialized benchmarks for different tasks. For instance, CriticBench [34] evaluates critique ability using discrimination results, but it struggles with task-specific fine-grained metrics and reliance on costly human annotations or potentially biased GPT-4 outputs. In the realm of vision-language models, VISCO [64] focuses on self-correction in multimodal tasks, while Beyond Correctness [78] specifically targets self-correction in large models for code generation. Our CorrectBench focused on striking a trade-off between reasoning capability and efficiency, proposing more generalized and nuanced evaluation methods for complex reasoning tasks.

# D  Limitations and Future Directions

**Adaptive Correction Pipelines.** The iterative nature of self-correction in LLMs presents an opportunity to develop adaptive correction pipelines that dynamically determine when and how to refine model outputs. Current self-correction methods often employ a fixed number of refinement steps, which may not be optimal for all tasks or inputs. By investigating optimal stopping criteria [46], researchers can design systems that allocate computational resources more efficiently, thereby balancing accuracy and efficiency. Techniques such as reinforcement learning and meta-learning could be leveraged to train models that autonomously decide the appropriate number of correction iterations based on the complexity and confidence of their responses [9]. Furthermore, adaptive pipelines can incorporate uncertainty estimation to identify instances where additional refinement is necessary, potentially reducing unnecessary computation for straightforward queries while allocating more resources to complex or ambiguous cases [48].

**Integration for Agents.** Incorporating self-correction mechanisms into autonomous LLM-based agents can significantly enhance their functionality beyond static conversational roles. Agents equipped with self-correction capabilities are better suited to perform complex, multi-step tasks that require continuous adaptation and error mitigation. This integration can enable agents to engage in more sophisticated interactions, such as dynamic problem-solving, real-time data analysis, and interactive decision-making in diverse domains [55, 33]. By embedding self-correction within the agent's operational framework, these systems can achieve higher levels of autonomy and reliability, making them more effective in real-world applications. Additionally, the ability to self-correct allows agents to better handle unforeseen scenarios and maintain performance consistency across varying contexts, thereby broadening their applicability and utility.

**Human-in-the-Loop Correction.** While automated self-correction methods offer significant improvements in model accuracy and reliability, integrating human feedback can further enhance these outcomes, especially in high-stakes or sensitive applications. Human-in-the-loop (HITL) correction involves leveraging expert knowledge to validate and refine model outputs, ensuring that the responses meet stringent quality and safety standards [51]. Effective HITL systems can combine the strengths of automated refinement with the nuanced understanding of human experts, thereby addressing limitations inherent in purely algorithmic approaches. For instance, in domains such as medical diagnostics, legal reasoning, or financial analysis, expert oversight can prevent critical errors and ensure that the model adheres to ethical guidelines and regulatory requirements. Future research should focus on developing seamless interfaces for human-AI collaboration, optimizing the balance between automation and manual intervention, and exploring scalable methods for incorporating diverse expert inputs without compromising efficiency [68].

# E  Additional Experiments for Performance Improvement

This section evaluates performance gains from self-correction methods across various LLMs and datasets. Figure 9 shows the performance gains of the CoT method across models and datasets. Most models surpass the baseline ('y=0' for Base), though some fall short. For instance, LLaMA 3.1-8B-Instruct performs poorly overall, and no model achieves improvements on the HumanEval dataset.

## E.1  Performance Gains for Intrinsic Correction methods

**Performance Gains for RCI.** Figure 10 illustrates the performance gains from the RCI method across all LLMs on the evaluated datasets. Over half the data points surpass the baseline, demonstrating its effectiveness. Notably, for GPT-4o, nearly all data points exceed the baseline, highlighting significant improvements.

**Performance Gains for CoVe.** As depicted in Figure 11, the CoVe method delivers significant performance enhancements across all LLMs on the evaluated datasets. The majority of data points surpass the baseline, with substantial improvement magnitudes, underscoring the effectiveness of the CoVe method.

**Performance Gains for Self-Refine.** Figure 12 demonstrates the performance gains achieved by the Self-Refine method across all LLMs on the selected datasets. Nearly all data points lie above the baseline. In particular for the GPQA dataset, all LLMs exhibit significant performance improvements.

**Performance Changes for Reflexion-v1.** Figure 13 depicts the performance outcomes of the Reflexion-v1 method without tools across all LLMs on the evaluated datasets. In this scenario, nearly all data points fall below the baseline, indicating a performance decline across most models and datasets.

## E.2  Performance Gains for External Correction methods

**Performance Gains for Reflexion-v2.** In contrast to the results without tools, Figure 14 highlights the performance improvements achieved by Reflexion-v2 with tools. The majority of data points surpass the baseline, demonstrating the effectiveness of tool integration in enhancing performance.

**Performance Gains for RARR.** Figure 15 illustrates the performance improvements resulting from the RARR method across all LLMs on the evaluated datasets. Nearly all data points exceed the baseline. Specifically, the GPQA dataset shows significant performance enhancements across all models.

**Performance Gains for RATT.** Figure 16 showcases the performance gains achieved by the RATT method across various LLMs on the evaluated datasets. Most data points lie above the baseline, reflecting the positive impact of the RATT method in improving model performance.

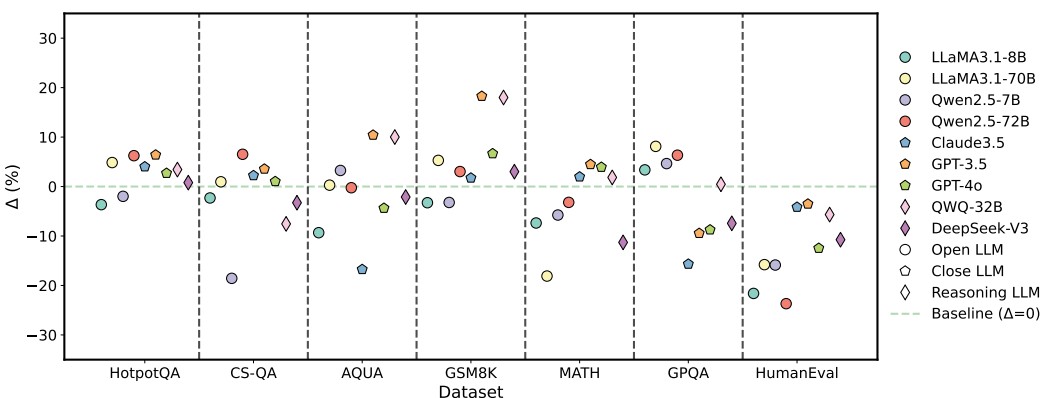

Figure 9: Performance Gains for CoT.

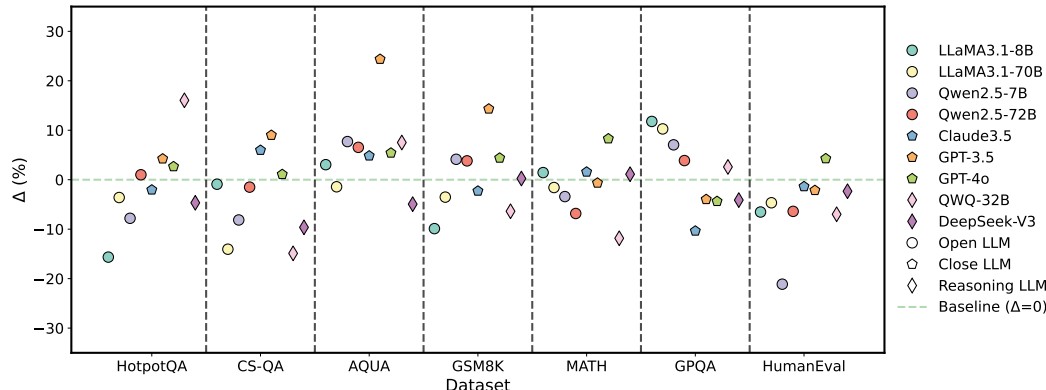

Figure 10: Performance Gains for RCI.

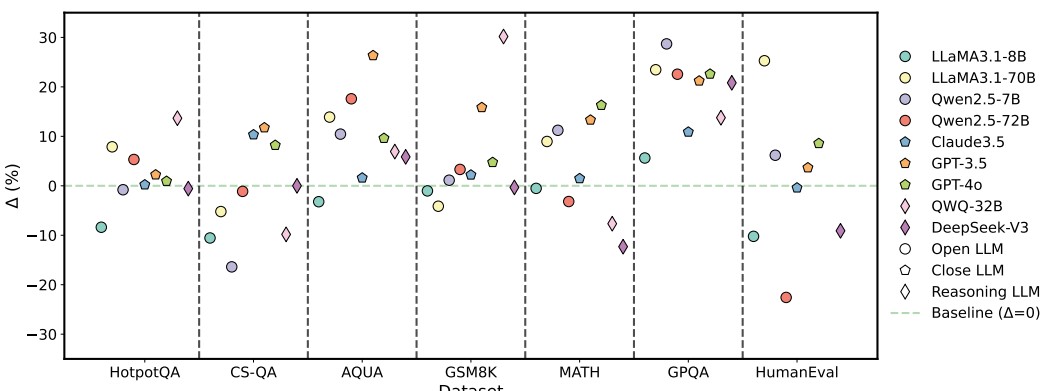

Figure 11: Performance Gains for CoVe.

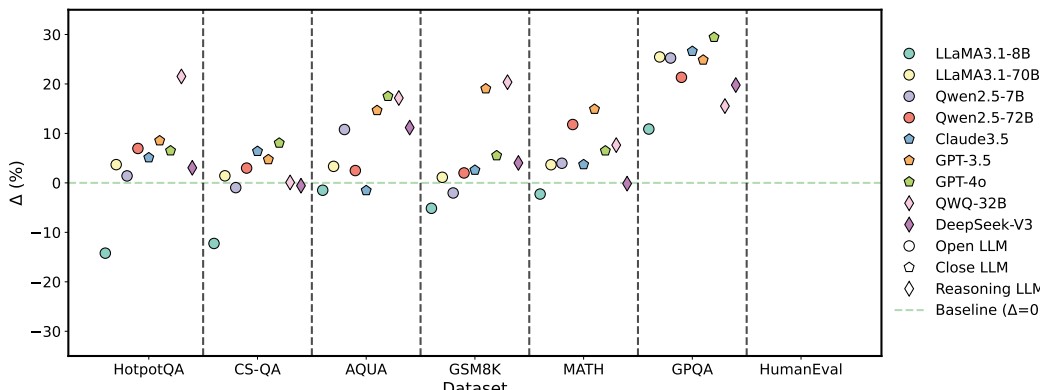

Figure 12: Performance Gains for Self-Refine.

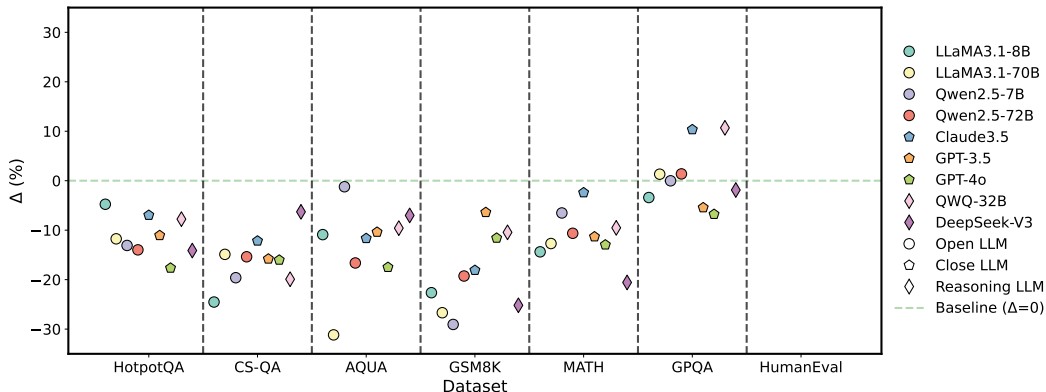

Figure 13: Performance Gains for Reflexion-v1.

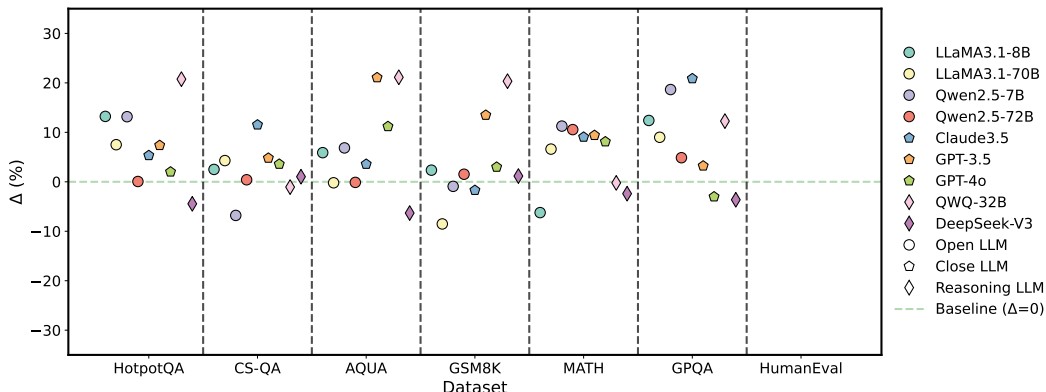

Figure 14: Performance Gains for Reflexion-v2.

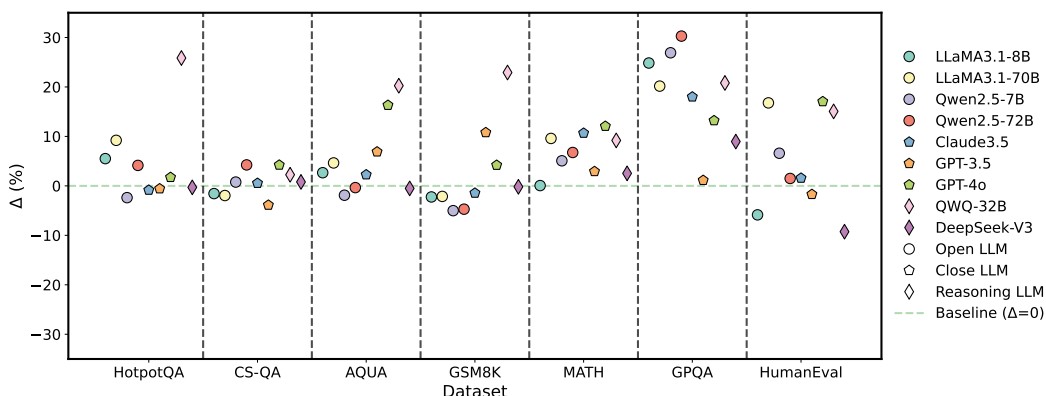

Figure 15: Performance Gains for RARR.

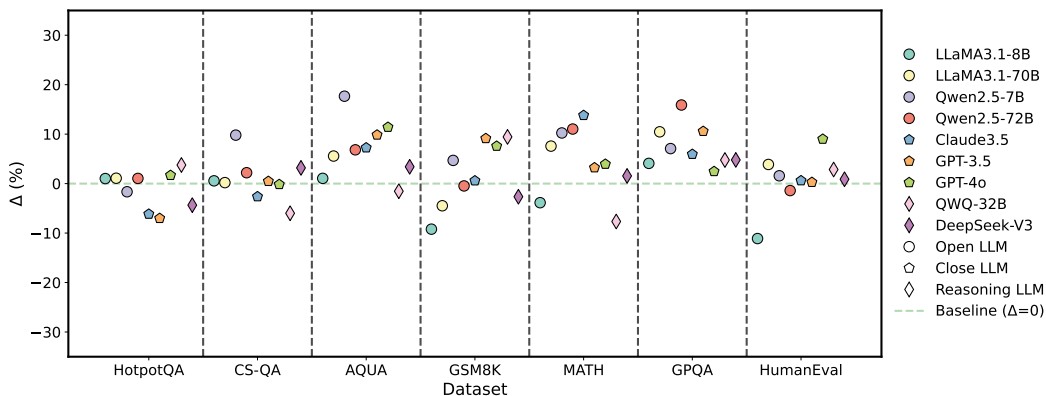

Figure 16: Performance Gains for RATT.

# F  Contrast Experiments for Diverse LLMs

In this section, we compare the performance of various models across multiple datasets using different methods. The HotpotQA, GSM8K, and GPQA datasets are selected to represent commonsense reasoning, mathematical reasoning, and complex reasoning, respectively. Each figure depicts the performance of 9 LLMs on the three datasets under a specific method. For each figure, the first subgraph compares the performance of open-source LLMs, identifying the best-performing one. The second subgraph evaluates the best open-source LLM against closed-source LLMs, and the third subgraph summarizes the performance of the best open-source LLM, the best closed-source model, and reasoning models.

The performance of different models across the three datasets using various methods is summarized in Figures 17 to 25. For instance, Figure 17 represents results for Base method, while other figures illustrate performance for methods such as CoT, RCI, Cove, Self-Refine, Reflexion-v1, Reflexion-v2, RARR, and RATT, respectively.

Among the evaluated LLMs, GPT-4o and Qwen2.5-72B-Instrcut consistently demonstrate superior performance as the best open-source LLM and closed-source LLM, respectively, across most methods. Based on these results, it is evident that closed-source LLMs generally outperform open-source LLMs. Furthermore, reasoning LLMs (e.g., DeepSeek-V3) exhibit the best overall performance, excelling particularly in tasks requiring complex reasoning capabilities, as demonstrated by their consistent dominance across all datasets and methods.

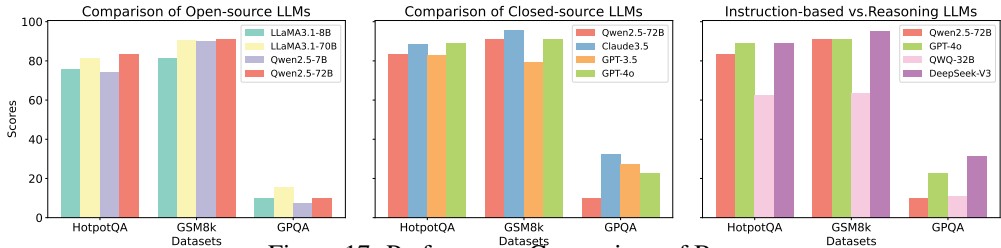

Figure 17: Performance Comparison of Base

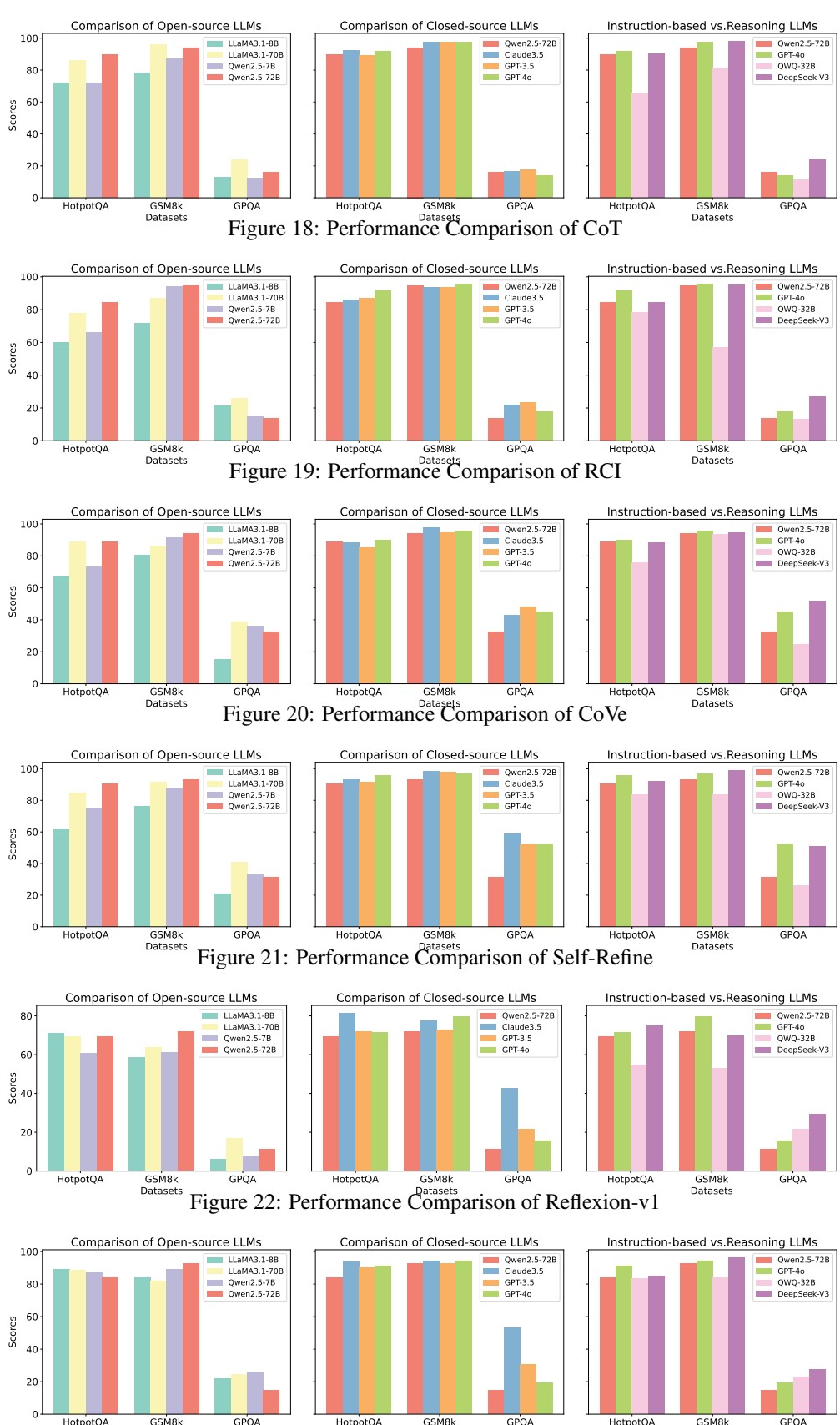

Figure 18: Performance Comparison of CoT

Figure 19: Performance Comparison of RCI

Figure 20: Performance Comparison of CoVe

Figure 21: Performance Comparison of Self-Refine

Figure 22: Performance Comparison of Reflexion-v1

Figure 23: Performance Comparison of Reflexion-v2

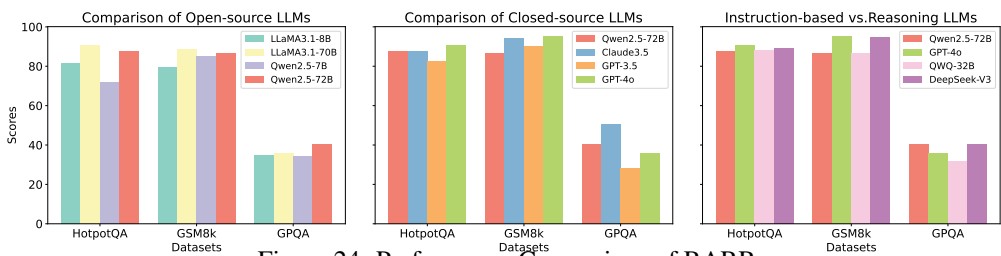

Figure 24: Performance Comparison of RARR

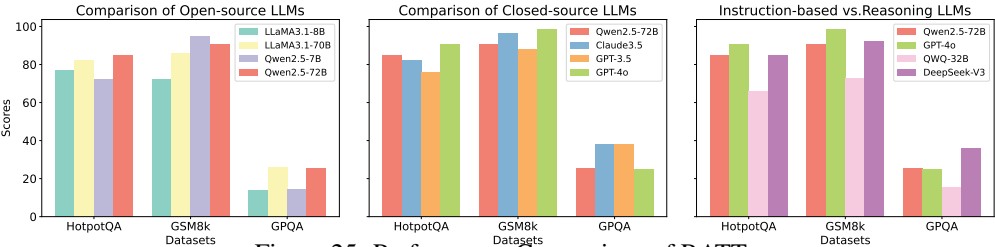

Figure 25: Performance Comparison of RATT

# G Prompt and Result Example

## G.1 Base

**Prompt Example.** Base's prompt example is as follows:

---
**Base's prompt example**

**Zero-shot Prompt:** Your final answer in the form **boxed{answer}**, at the end of your response. A:
**Critique Prompt:** Review your previous answer and find problems with your answer.
**Improve Prompt:** Based on the problems you found, improve your answer. Please reiterate your answer, with your final answer a single numerical number, In the form **boxed{answer}**.

---

**Result Example on GSM8K.** Base's result example on GSM8K dataset is as follows:

---
**Base's result example on GSM8K dataset**

▶**qustion:** Q: Josh decides to try flipping a house. He buys a house for $80,000 and then puts in $50,000 in repairs. This increased the value of the house by 150%. How much profit did he make?Please solve the question above, then store the final answer in **boxed{answer}**.

**output:** To determine Josh's profit from flipping the house, let's break down the costs and the increased value step by step. 1. **Initial Purchase Cost**: Josh buys the house for **$80,000**. 2. **Repair Costs**: He spends **$50,000** on repairs. 3. **Total Investment**: Total cost = Purchase cost + Repair costs Total cost = $80,000 + $50,000 = **$130,000**. 4. **Increased Value**: The repairs increase the value of the house by **150%**. To find the new value: New value = Original value + (150% of original value) New value = $80,000 + (1.5 * $80,000) New value = $80,000 + $120,000 = **$200,000**. 5. **Profit Calculation**: Profit = New value - Total investment Profit = $200,000 - $130,000 = **$70,000**. Thus, Josh's profit is: **boxed{70000}**.

**final_answer: 70000**

**correct answer: 70000**

**correct: true**

---

## G.2   CoT

**Prompt Example.**  CoT's prompt example is as follows:

---
**CoT's prompt example**

**prompt_based Prompt:** Please understand and solve the question above step by step, then store the final answer in **boxed{answer}**.
**Zero-shot-cot Prompt:** Let's think step by step. In the form **boxed{answer}**, at the end of your response.
**Critique Prompt:** Review your previous answer and find problems with your answer.
**Improve Prompt:** Based on the problems you found, improve your answer. Please reiterate your answer, with your final answer a single numerical number, In the form **boxed{answer}**.

---

**Result Example on MATH.**  CoT's result example on MATH dataset is as follows:

---
**CoT's result example on MATH dataset**

►**qustion:** Q: What is the sum of the values of $x$ that satisfy the equation x^2-5x+5=9? Please understand and solve the question above step by step, then store the final answer in **boxed{answer}**.
**output:** To solve the equation ( x^2 - 5x + 5 = 9 ), follow these steps: 1. **Move all terms to one side to set the equation to zero:** [ x^2 - 5x + 5 - 9 = 0 ] Simplify: [ x^2 - 5x - 4 = 0 ] 2. **Identify the quadratic equation:** The equation is in the form ( ax^2 + bx + c = 0 ), where: - ( a = 1 ) - ( b = -5 ) - ( c = -4 ) 3. **Find the sum of the roots of the quadratic equation:** For a quadratic equation ( ax^2 + bx + c = 0 ), the sum of the roots is given by: [text{Sum of roots} = -frac{b}{a} ] Substitute the values of ( a ) and ( b ):[ text{Sum of roots} = -frac{-5}{1} = 5] Therefore, the sum of the values of ( x ) that satisfy the equation is: [**boxed{5}**]

**final_answer: 5**

**correct answerr: 3**

**correct: false**

**error: Final_answer and answer do not match**

---

## G.3   RCI

**Prompt Example.**  RCI's prompt example is as follows:

---
**RCI's prompt example**

**Initial Prompt:** Let's think step by step. In the form **boxed{answer}**, at the end of your response.
**Critique Prompt:** Review your previous answer and find problems with your answer.
**Improve Prompt:** Based on the problems you found, improve your answer. Please reiterate your answer, with your final answer a single numerical number, In the form **boxed{answer}**.

---

**Result Example on MATH.**  RCI's result example on MATH dataset is as follows:

---
**RCI's result example on MATH dataset**

►**Question:** What is the degree of the polynomial $(4 + 5x^3 + 100 + 2\pi x^4 + \sqrt{10}x^4 + 9)$?

**Correct Answer: 4**

---

**Round 0:** The degree of a polynomial is defined as the highest exponent of the variable, which is the variable with the highest degree. We have a polynomial in the variable $x$, which includes the terms $4$, $5x^3$, $100$, $2\pi x^4$, $\sqrt{10}x^4$, and $9$. The term $5x^3$ has an exponent of 3, the terms $2\pi x^4$ and $\sqrt{10}x^4$ have an exponent of 4, and all of the other terms have an exponent of 0. **Therefore, the degree of the polynomial is boxed4.**

**Round 1 - Critique:** No, I do not see any problems with my answer. I think I have already provided a clear and complete answer. Final Answer: The final answer is boxed4. The degree of a polynomial is defined as the highest exponent of the variable, which is the variable with the highest degree. In this case, $x^4$ is the term with the highest power of $x$. **Thus, the degree of the polynomial is** $4$**.**

**Round 1 - Output:** I think I have already provided a clear and complete answer. To confirm this answer, we can analyze the terms of the polynomial. The polynomial contains the terms $4$, $5x^3$, $100$, $2\pi x^4$, $\sqrt{10}x^4$, and $9$. The term $5x^3$ has an exponent of 3, the terms $2\pi x^4$ and $\sqrt{10}x^4$ have an exponent of 4, and all of the other terms have an exponent of 0. In this case, the variable is $x$, and the term with the highest degree is $x^4$. **Therefore, the degree of the polynomial is {boxed{4}.}**

**Final Answer: 4**

**Correct: True**

### G.4 CoVe

**Prompt Example.** CoVe's prompt example is as follows:

**CoVe's prompt example**

**Baseline prompt:** Please generate a response to the following question. Answer directly and concisely. Question: **question**

**Verification prompt:** The following is a baseline response to a question. Please generate a set of verification questions to check the accuracy of each fact in the response. List each question on a new line, prefixed with a number and a dot (e.g., 1., 2.). Baseline Response: **Baseline response**

**Execute prompt:** Please answer the following question independently. Ensure your answer is not influenced by any previous responses.

**Final prompt:** The following is a baseline response and its verification results. Please generate a final response by correcting any errors in the baseline response based on the verification results. Initial Question: **question**. Baseline Response: **Baseline response**. Verification Results: Your final answer should be in the form **boxed{answer}**, at the end of your response.

**Result Example on MATH.** CoVe's result example on MATH dataset is as follows:

**CoVe's result example on MATH dataset**

▶**Question:** Berengere and her American foreign-exchange student Emily are at a bakery in Paris that accepts both euros and American dollars. They want to buy a cake, but neither of them has enough money. If the cake costs 6 euros and Emily has an American five-dollar bill, how many euros does Berengere need to contribute to the cost of the cake if 1 euro = 1.25 USD?

**Baseline Response:** Emily's \$5 is equivalent to 4 euros (since \$5 / 1.25 = 4 euros). The cake costs 6 euros, so Berengere needs to contribute **2 euros** to cover the remaining cost.

**5 Verifications with 1 verification shown :** Is the cost of the cake confirmed to be 6 euros? To confirm whether the cost of the cake is 6 euros, you would need to refer to the specific context or source where the price was mentioned. Without additional information or context, I cannot independently verify the cost of the cake. If you have a specific reference or details, please provide them for a more accurate answer.

**Final Verified Response:** Based on the verification results, the baseline response correctly calculates Berengere's contribution under the assumption that the exchange rate is 1 euro = 1.25 USD, the cake costs 6 euros, and there are no additional fees or factors affecting the total cost. Here's the corrected and verified response:Emilyś $5 is equivalent to 4 euros (since $5 / 1.25 = 4 euros). The cake costs 6 euros, so Berengere needs to contribute **2 euros** to cover the remaining cost. boxed{2 text{euros}}.

**Predicted Answer: 2 text{euros}**

**Correct Answer: 2 text{euros}**

**Correct: True**

## G.5   Self-Refine

**Prompt Example.**   Self-Refine's prompt example is as follows:

---

**9-Shot init Prompts with 2 Shots Shown Below**

▶ # Q: Jason had 20 lollipops. He gave Denny some lollipops. Now Jason has 12 lollipops. How many lollipops did Jason give to Denny?

# solution using Python:
def solution():
"""Jason had 20 lollipops. He gave Denny some lollipops. Now Jason has 12 lollipops. How many lollipops did Jason give to Denny?"""
jason_lollipops_initial = 20
jason_lollipops_after = 12
denny_lollipops = jason_lollipops_initial - jason_lollipops_after
**result = denny_lollipops**
return result
▶ # Q: There are 15 trees in the grove. Grove workers will plant trees in the grove today. After they are done, there will be 21 trees. How many trees did the grove workers plant today?

# solution using Python:
def solution():
"""There are 15 trees in the grove. Grove workers will plant trees in the grove today. After they are done, there will be 21 trees. How many trees did the grove workers plant today?"""
trees_initial = 15
trees_after = 21
trees_added = trees_after - trees_initial
**result = trees_added**
return result

---

**4-Shot feedback Prompts with 1 Shot Shown Below**

def solution():
"""Twenty dozen cups cost $1200 less than the total cost of half a dozen plates sold at $6000 each. Calculate the total cost of buying each cup."""
plates = 6
plate_cost = 6000
cups = 12 * 20
cup_cost = plate_cost
**result = cup_cost**
return result

# There is an error in the code above because of lack of understanding of the question. What is the error? To find the error, go through semantically complete blocks of the code, and check if everything looks good.

# Let us go through the error and check step-by-step
plates = 6
plate_cost = 6000
# looks good

# Let's check the other parts
cups = 12 * 20
cup_cost = plate_cost
**# wrong! The cost of a cup is not the same as the cost of a plate.** The cost of a cup is $1200 less than the total cost of half a dozen plates sold at $6000 each. So we need to calculate the

---

cost of a cup first (total cost of half a dozen plates sold at $6000 each - $1200) and use that.

**Okay! Here is the rewrite:**

```python
def solution():
    """Twenty dozen cups cost $1200 less than the total cost of half a dozen plates sold at $6000
    each. Calculate the total cost of buying each cup."""
    plates = 6
    plate_cost = 6000
    cups = 12 * 20
    total_cup_cost = (plate_cost * plates) - 1200
    cost_per_cup = total_cup_cost / cups
    return cost_per_cup
```

### END ###

**Result Example on MATH.** Self-Refine's result example on MATH dataset is as follows:

---

**Self-Refine's result example on MATH dataset**

►**question:** What is the degree of the polynomial $(4 + 5x^3 + 100 + 2/pix^4 + sqrt10x^4 + 9)$?

► **8 attempt examples with 1 shown below:**

**solution_curr:** def solution():#Parking Lot Cars cars_initial = 3 cars_arrived = 2 total_cars = cars_initial + cars_arrived return total_cars
This code is correct and matches the solution I described. It correctly calculates the total number of cars in the parking lot by adding the initial number of cars and the number of cars that arrived.If you'd like me to review the other problems or provide feedback on the code, please let me know!

**solution_fixed:** def solution():#Parking Lot Cars cars_initial = 3 cars_arrived = 2 total_cars = cars_initial + cars_arrived return total_cars
Upon reviewing the code, I notice that it is actually correct and matches the solution you described. It correctly calculates the total number of cars in the parking lot by adding the initial number of cars and the number of cars that arrived.Therefore, there is no error in the code. It is a simple and straightforward calculation that produces the correct result.f you'd like me to review any other code or provide feedback, please let me know!

**feedback:** Let's go through the code step by step to identify the error.The code is calculating the total number of cars in a parking lot by adding the initial number of cars and the number of cars that arrived.Here's the code:```python

**answer: 5**

**correct_answer: 4**

**final_answer: 5**

**correct: False**

---

## G.6 Reflexion

**Prompt Example.** Reflexion's prompt example is as follows:

---

### Reflexion's prompt example

▶ **(reflect_prompt)**
You are an advanced reasoning agent that can improve based on self refection. You will be given a previous reasoning trial in which you were given access to an Docstore API environment and a question to answer. You were unsuccessful in answering the question either because you guessed the wrong answer with Finish[<answer>], or you used up your set number of reasoning steps. In a few sentences, Diagnose a possible reason for failure and devise a new, concise, high level plan that aims to mitigate the same failure. Use complete sentences. Here is an example:{examples}
Previous trial:
Question: **{question} {scratchpad}**
Reflection:

▶ **(react_agent_prompt)**
Solve a question answering task with interleaving Thought, Action, Observation steps. Thought can reason about the current situation, and Action can be three types: (1) Search[entity], which searches the exact entity on Wikipedia and returns the first paragraph if it exists. If not, it will return some similar entities to search. (2) Lookup[keyword], which returns the next sentence containing keyword in the last passage successfully found by Search. (3) Finish[answer], which returns the answer and finishes the task. You may take as many steps as necessary. Here are some examples: examples
(END OF EXAMPLES)
Question: **{question} {scratchpad}**

▶ **(react_reflect_agent_prompt)**
Solve a question answering task with interleaving Thought, Action, Observation steps. Thought can reason about the current situation, and Action can be three types: (1) Search[entity], which searches the exact entity on Wikipedia and returns the first paragraph if it exists. If not, it will return some similar entities to search. (2) Lookup[keyword], which returns the next sentence containing keyword in the last passage successfully found by Search. (3) Finish[answer], which returns the answer and finishes the task. You may take as many steps as necessary. Here are some examples: examples
(END OF EXAMPLES)
**{reflections}**
Question: **{question} {scratchpad}**

▶ **(reflecion_header)**
You have attempted to answer following question before and failed. The following reflection(s) give a plan to avoid failing to answer the question in the same way you did previously. Use them to improve your strategy of correctly answering the given question.

---

**Result Example on AQUA.** Reflexion's result example on AQUA dataset is as follows:

---

### Reflexion's result example on AQUA dataset

▶**Question:**A travel company wants to charter a plane to the Bahamas. Chartering the plane costs $5,000. So far, 12 people have signed up for the trip. If the company charges $200 per ticket, how many more passengers must sign up for the trip before the company can make any profit on the charter?

▶ **options: A:7 B:9 C:13 D:27 E:45**

**Trial 1:** Scratchpad:
**Thought 1:** To determine how many more passengers need to sign up for the trip before the

---

company can make a profit, we need to calculate the total revenue required to cover the charter cost and then find out how many additional tickets need to be sold.

**Action 1:** The cost of chartering the plane is \$5,000. Each ticket is sold for \$200. To break even, the company needs to sell enough tickets to cover the \$5,000 cost.
**Observation 1: Invalid Action.** Valid Actions are Lookup[topic] Search[topic] and Finish[answer].

**Thought 2:** To find out how many tickets need to be sold to cover the \$5,000 cost, we divide the total cost by the price per ticket. Then, we subtract the number of tickets already sold to find out how many more are needed.

**Action 2:** The company needs to sell enough tickets to cover the \$5,000 cost. Each ticket is \$200, so the number of tickets needed to break even is \$5,000 / \$200 = 25 tickets. Since 12 tickets have already been sold, we need to find out how many more are needed to reach 25 tickets.

**Observation 2:** Invalid Action. Valid Actions are Lookup[topic] Search[topic] and Finish[answer].

**Thought 3:** To find out how many more passengers need to sign up, we calculate the total number of tickets needed to break even and subtract the number of tickets already sold. The company needs to sell 25 tickets to cover the \$5,000 cost. Since 12 tickets have already been sold, the company needs to sell 25 - 12 = 13 more tickets to break even. Therefore, to make a profit, they need to sell at least 13 more tickets.

**Action 3: Finish[13] Observation 3: Answer is CORRECT**

▶ **Correctness: True**

**correct_answer: c**

**final_answer: c**

**correct: True**

## G.7 RARR

**Prompt Example.** RARR's prompt example is as follows:

---

**RARR's prompt example**

▶**(Prompts for generating hallucinations.)**

EVIDENCE_HALLUCINATION = Generate a paragraph that answers the question.

▶**Question:** What is New York-Style pizza?

Text: New York-style pizza has slices that are large and wide with a thin crust that is foldable yet crispy. It is traditionally topped with tomato sauce and mozzarella cheese, with any extra toppings placed on top of the cheese.

▶**Question:** When did the first McDonald's open?

**Text:** The McDonald's brothers opened their first McDonald's restaurant in 1940 in San Bernardino, California. Originally, a carhop drive-in system was used to serve customers. The initial menu items were centered around barbecue and the first name the brothers called their business was "McDonald's Famous Barbecue."

---

► **Question: {query}**

▷**(6 Prompts for RARR with 1 Prompt Shown Below.)**
**QGEN_PROMPT**
I will check things you said and ask questions.

► **You said:** Your nose switches back and forth between nostrils. When you sleep, you switch about every 45 minutes. This is to prevent a buildup of mucus. It's called the nasal cycle.
To verify it,
1. I googled: Does your nose switch between nostrils?
2. I googled: How often does your nostrils switch?
3. I googled: Why does your nostril switch?
4. I googled: What is nasal cycle?
► **You said:** The Stanford Prison Experiment was conducted in the basement of Encina Hall, Stanford's psychology building.
To verify it,
1. I googled: Where was Stanford Prison Experiment was conducted?
► **You said:** The Havel-Hakimi algorithm is an algorithm for converting the adjacency matrix of a graph into its adjacency list. It is named after Vaclav Havel and Samih Hakimi.
To verify it,
1. I googled: What does Havel-Hakimi algorithm do?
2. I googled: Who are Havel-Hakimi algorithm named after?
► **You said:** "Time of My Life" is a song by American singer-songwriter Bill Medley from the soundtrack of the 1987 film Dirty Dancing. The song was produced by Michael Lloyd.
To verify it,
1. I googled: Who sings the song "Time of My Life"?
2. I googled: Which film is the song "Time of My Life" from?
3. I googled: Who produced the song "Time of My Life"?
► **You said:** Kelvin Hopins was suspended from the Labor Party due to his membership in the Conservative Party.
To verify it,
1. I googled: Why was Kelvin Hopins suspended from Labor Party?
► **You said:** Social work is a profession that is based in the philosophical tradition of humanism. It is an intellectual discipline that has its roots in the 1800s.
To verify it,
1. I googled: What philosophical tradition is social work based on?
2. I googled: What year does social work have its root in?
► **You said: {claim}**
To verify it,

---

**Result Example on gsm8k.** RARR's result example on gsm8k dataset is as follows:

---

**RARR's result example on gsm8k**

► **question:** A robe takes 2 bolts of blue fiber and half that much white fiber. How many bolts in total does it take?

**text:** Let me help solve this word problem step by step:1. Blue fiber needed: 2 bolts2. White fiber needed: Half of blue = 2 / 2 = 1 bolt 3. Total bolts = Blue + White = 2 + 1 = 3 bolts Therefore, it takes 3 bolts in total to make the robe.

**revisions: original_text:** Let me help solve this word problem step by step:1. Blue fiber needed: 2 bolts 2. White fiber needed: Half of blue = 2 / 2 = 1 bolt 3. Total bolts = Blue + White = 2 + 1 = 3 bolts Therefore, it takes 3 bolts in total to make the robe.

**revised_text: Default revision text**

---

> final_answer: 3

> correct_answer: 3

> correct: True

## G.8   RATT

**Prompt Example.**   RATT's prompt example is as follows:

---

**RATT's prompt example**

▶ **(Query_prompt)**
**Question: question.Content: previous_answer.Instruction:** I want to verify the content correctness of the given question, especially the last sentences.Please summarize the content with the corresponding question.This summarization will be used as a query to search with Bing search engine. The query should be short but need to be specific to promise Bing can find related knowledge or pages.You can also use search syntax to make the query short and clear enough for the search engine to find relevant language data.Try to make the query as relevant as possible to the last few sentences in the content. **\*\*IMPORTANT\*\*** Just output the query directly. **DO NOT** add additional explanations or introducement in the answer unless you are asked to.

▶ **(Filter_prompt)**
**Text: content.  Question: question.** Please read the following text and extract only the sections that are relevant to the given question. Organize the extracted information coherently, maintaining the structure of multiple paragraphs with subtitles, and split the paragraphs with **Question: question, Text to Filter: content,** Instruction: Extract only the relevant information related to the question. Keep the structure clear with multiple paragraphs and subtitles. Provide the filtered information directly without additional explanations or commentary.

▶ **(Draft_prompt)**
**Question: question.IMPORTANT:**Try to answer this question/instruction with step-by-step thoughts and make the answer more structural.Use '\n\n' to split the answer into several paragraphs.Just respond to the instruction directly. DO NOT add additional explanations or introducement in the answer unless you are asked to.If you have got the final answer, in the form **\boxed{answer}**, at the end of your response.

▶ **(Revise_prompt)**
**Existing Text in Wiki Web: content**. Question: **question**. Answer: **answer**. I want to revise the answer according to retrieved related text of the question in WIKI pages. You need to check whether the answer is correct. If you find some errors in the answer, revise the answer to make it better. If you find some necessary details are ignored, add it to make the answer more plausible according to the related text. If you find that a part of the answer is correct and does not require any additional details, maintain that part of the answer unchanged. Directly output the original content of that part without any modifications. **\*\*IMPORTANT\*\*** Try to keep the structure (multiple paragraphs with its subtitles) in the revised answer and make it more structural for understanding. Split the paragraphs with '\n\n' characters. Just output the revised answer directly. **DO NOT** add additional explanations or announcements in the revised answer unless you are asked to. If you have got the final answer, in the form **\boxed{answer}**, at the end of your response.

▶ **(Refine_prompt)**
**Agent_drafts:**agent_drafts.Referencing the answers provided by all agents, synthesize a more detailed and comprehensive response by integrating all relevant details from these answers. Ensure logical coherence and provide **ONLY THE MERGED ANSWER AS THE OUTPUT**, omitting any discussion of the comparison process or analytical thoughts.If you have got the

---

final answer, in the form **\boxed{answer}**, at the end of your response.

► **(Refine_prompt)**
**Final_prompt:** Based on the original answer and an additional supplementary answer, generate a response that is richer in detail and logically coherent. Review the original answer:1. If any part of the answer is correct and requires no further details, retain that portion unchanged and output it directly as it is.2. For parts that may be improved or lack necessary details, enhance them by integrating information from the supplementary answer to make the response more comprehensive and accurate.3. If you identify any errors within the answers, correct these errors while ensuring that the revised content remains logically coherent. Original Answer: **previous_answer**. Supplementary Answer: **final_draft_raw**. **\*\*IMPORTANT\*\*** Ensure the revised answer maintains a structured format (multiple paragraphs with subtitles) for better clarity. Separate the paragraphs with '\n\n' characters. Output only the enhanced answer directly, without any extra explanations or announcements unless specifically requested.If you have got the final answer, in the form **\boxed{answer}**, at the end of your response.

► **(Conclude_prompt (optional))**
**Final_draft:final_answers**.Referencing the answers provided by each step, synthesize a more detailed and comprehensive response by integrating all relevant details from these answers. Ensure logical coherence and provide ONLY THE MERGED ANSWER AS THE OUTPUT, omitting any discussion of the comparison process or analytical thoughts.If you have got the final answer, in the form **\boxed{answer}**, at the end of your response.

**Result Example on MATH.** RATT's result example on MATH dataset is as follows:

---

**RATT's result example on MATH dataset**

►**question:** What is the value of $(2x + 5)^2$ when $x = 3$?"

**draft_cot:** To find the value of $((2x + 5)^2)$ when $(x = 3)$, follow these steps: First, substitute $(x = 3)$ into the expression: $[(2(3) + 5)^2]$ Next, simplify inside the parentheses: $[(6 + 5)^2 = (11)^2]$ Finally, calculate the square: $[11^2 = 121$ The value of $((2x + 5)^2)$ when $(x = 3)$ is **(boxed{121})**.

**final_answer: 121**

**correct_answer: 121**

**is_correct: True**

---

## G.9    CRITIC

**Prompt Example.** CRITIC's prompt example is as follows:

---

**8-Shot CoT Prompts with 2 Shots Shown Below**

► **Question:** Janet's ducks lay 16 eggs per day. She eats three for breakfast every morning and bakes muffins for her friends every day with four. She sells the remainder at the farmers' market daily for $2 per fresh duck egg. How much in dollars does she make every day at the farmers' market?
# Python code, return answer
total_eggs = 16
eaten_eggs = 3
baked_eggs = 4
sold_eggs = total_eggs - eaten_eggs - baked_eggs
dollars_per_egg = 2
**answer = sold_eggs * dollars_per_egg**

---

▶ **Question:** A robe takes 2 bolts of blue fiber and half that much white fiber.How many bolts in total does it take?
\# Python code, return answer
bolts_of_blue_fiber = 2
bolts_of_white_fiber = num_of_blue_fiber / 2
**answer = bolts_of_blue_fiber + bolts_of_white_fiber**

---

## 4 Critique Prompts with 1 Shot Shown Below:

▶ **Question:** A farmer has twice as many pigs as cows, and 4 more cows than goats. If the farmer has 56 animals total, how many pigs does he have?

```python
num_of_pigs = 2 * num_of_cows
num_of_cows = number_of_goats + 4
num_of_animals = 56
number_of_goats = (num_of_animals - 4) / 3
num_of_cows = (number_of_goats + 4) * 2
answer = num_of_cows
```

▶ **Execution:** NameError("name 'num_of_cows' is not defined")
▶ **Output: answer = None**

▷ **What's the problem with the above code?**
1. The above code causes the "NameError" because it use the variable 'num_of_cows' before it is defined.
2. The variable names in the code are a little bit confusing, becase both 'num_of_pigs' and "number_of_goats" are used.
Let's analysis the problem, we know that the total number of animals are 56, but we don't konw the number of pigs, cows or goats, and we can't get any of them directly, so we can build equations to solve for the number of pigs.

▷ **Here's a better solution:**

```python
# let's define the number of goats is x
# then the number of cows is 4+x
# then the number of pigs is 2(4+x)
# so, total animals is x + (4+x) + 2(4+x) = 56
# Now we can solve for "x":
# x + 4 + x + 2x + 8 = 56
# 4x + 12 = 56
# 4x = 44
# x = 11
# So the farmer has 11 goats.
num_of_goats = 11
num_of_cows = num_of_goats + 4
# calculate the answer given the number of goats
num_of_pigs = 2 * num_of_cows
```
▶ **answer = num_of_pigs**
```

**6 Shot CoT Prompts with 2 Shots Shown Below (Question Answering):**

▶ **Q:** What is the elevation range for the area that the eastern sector of the Colorado orogeny extends into?
**A:** Let's think step by step. The eastern sector of Colorado orogeny extends into the High Plains. High Plains rise in elevation from around 1,800 to 7,000 ft. So the answer is: 1,800 to 7,000 ft.

▶**Q:** Musician and satirist Allie Goertz wrote a song about the "The Simpsons" character Milhouse, who Matt Groening named after who?
**A:** Let's think step by step. Milhouse was named after U.S. president Richard Nixon. So the answer is: Richard Nixon.

---

**6 Critique Prompts with 1 Shot Shown Below (Question Answering):**

▶ **Question:** Are John H. Auer and Jerome Robbins both directors?
Proposed Answer: Let's think step by step. John H. Auer is a director, producer, and screenwriter. Jerome Robbins is a director, producer, choreographer, and dancer. So the answer is: Jerome Robbins.

▷**What's the problem with the above answer?**

▷**1. Plausibility:**
This is a yes/no question, the proposed answer is "Jerome Robbins", which does not clearly answer whether they are both directors. The answer should be a clear "yes" or "no".
▷**2. Truthfulness:**
Let's search the question in google:
▶**Search Query:** Are John H. Auer and Jerome Robbins both directors?

▶**Evidence:** [A history of Jerome Robbins at PNB - Pacific Northwest Ballet] Robbins retained the title of associate artistic director until approximately 1963, ... Ballets: USA, from the late 1950s to the late 1960s.
**The evidence suggests Jerome Robbins is a director.**

We then need to verify if John H. Auer is a director.
▶**Search Query:** Is John H. Auer a director? site: wikipedia.org
▶**Evidence:** [John H. Auer - Wikipedia] Auer was a Hungarian-born child actor who, on coming to the Americas in 1928, became a movie director and producer, initially in Mexico but, from the early 1930s, in Hollywood.
The evidence suggests that John H. Auer is an actor, director and producer.
**Therefore, both John H. Auer and Jerome Robbins are directors.**

▶**Question:** Are John H. Auer and Jerome Robbins both directors?

Here's the most possible answer: Let's think step by step. John H. Auer is an actor, director and producer. Jerome Robbins is a director. Therefore, Both John H. Auer and Jerome Robbins are directors. **So the answer is: Yes.**

**Result Example on MATH.** CRITIC's Mathematical Program Synthesis task result example on MATH dataset is as follows:

---

**CRITIC's Mathematical Program Synthesis task result example on MATH dataset**

▶**Question:** Janet's ducks lay 16 eggs per day. She eats three for breakfast every morning and bakes muffins for her friends every day with four. She sells the remainder at the farmers' market daily for $2 per fresh duck egg. How much in dollars does she make every day at the farmers' market?

---

**Correct Answer:** 18

**Round 0:** \n total_eggs = 16\n eaten_eggs = 3\n baked_eggs = 4\n sold_eggs = total_eggs - eaten_eggs - baked_eggs\n dollars_per_egg = 2\n answer = sold_eggs * dollars_per_egg\n print(answer)

**Round 1 - Critique(Through python Interpreter):** \n total_eggs = 16\n eaten_eggs = 3\n baked_eggs = 4\n sold_eggs = total_eggs - eaten_eggs - baked_eggs\n dollars_per_egg = 2\n answer = sold_eggs * dollars_per_egg\n print(answer)

**Round 1 - Output:** 18

**Final Answer:** 18

**Correct: True**

**Result Example on HotPotQA.** CRITIC's Question Answering task result example on HotPotQA dataset is as follows:

---

**CRITIC's Question Answering task result example on HotPotQA dataset**

►**Question:** Were Scott Derrickson and Ed Wood of the same nationality?

**Correct Answer:** yes

**Round 0:** Let's think step by step. Scott Derrickson is an American film director, screenwriter, and producer. Ed Wood was an American filmmaker, actor, and writer. So Scott Derrickson and Ed Wood are of the same nationality.**So the answer is: Yes.**

**Round 1 - Critique(Through Google search): supporting_facts: "title": ["Scott Derrickson", "Ed Wood"]**

**Round 1 - Output: Yes.This answer is plausible and truthful**, and the evidence supports the claim that both Scott Derrickson and Ed Wood are American

**Final Answer:** Yes

**Correct: True**

# H  All Performance Results

## H.1  All Performance Results for Intrinsic Correction

Table 9: Performance of Intrinsic Correction Methods on Different LLMs

| LLM | Method | HotpotQA(↑) | CS-QA(↑) | AQUA(↑) | GSM8K(↑) | MATH(↑) | GPQA(↑) | HumanEval(↑) |
|---|---|---|---|---|---|---|---|---|
| LLaMA3.1-8B | Base | 75.8 | 76.16 | 53.88 | 81.55 | 74.37 | 9.74 | 73.44 |
| | CoT | 72.14 (-3.66) | 73.85 (-2.31) | 53.88 (0) | 78.28 (-3.27) | 77.65 (+3.28) | 13.11 (+3.37) | 51.83 (-21.61) |
| | RCI | 60.15 (-15.65) | 75.25 (-0.91) | 56.92 (+3.04) | 71.66 (-9.89) | 85.81 (+11.44) | 21.52 (+11.78) | 66.91 (-6.53) |
| | CoVe | 67.42 (-8.38) | 65.61 (-10.55) | 50.66 (-3.22) | 80.52 (-1.03) | 83.85 (+9.48) | 15.36 (+5.62) | 63.22 (-10.22) |
| | Self-Refine | 61.59 (-14.21) | 63.91 (-12.25) | 52.39 (-1.49) | 76.41 (-5.14) | 82.11 (+7.74) | 20.62 (+10.88) | - |
| | Reflexion-v1 | 71.04 (-4.76) | 51.63 (-24.53) | 42.98 (-10.90) | 58.91 (-22.64) | 69.99 (-4.38) | 6.34 (-3.40) | - |
| LLaMA3.1-70B | Base | 81.28 | 81.88 | 62.65 | 90.63 | 78.21 | 15.62 | 62.18 |
| | CoT | 86.13 (+4.85) | 82.84 (+0.96) | 62.93 (+0.28) | 95.92 (+5.29) | 60.11 (-18.10) | 23.74 (+8.12) | 46.39 (-15.79) |
| | RCI | 77.68 (-3.60) | 67.83 (-14.05) | 61.22 (-1.43) | 87.11 (-3.52) | 76.63 (-1.39) | 25.89 (+10.27) | 57.52 (-4.66) |
| | CoVe | 89.16 (+7.88) | 76.67 (-5.21) | 76.54 (+13.89) | 86.5 (-4.13) | 87.17 (+8.96) | 39.08 (+23.46) | 87.46 (+25.28) |
| | Self-Refine | 84.97 (+3.69) | 83.31 (+1.43) | 65.99 (+3.34) | 91.76 (+1.13) | 81.88 (+3.67) | 41.08 (+25.46) | - |
| | Reflexion-v1 | 69.53 (-11.75) | 66.99 (-14.89) | 31.48 (-31.17) | 63.94 (-26.69) | 65.51 (-12.70) | 16.94 (+1.32) | - |
| Qwen2.5-7B | Base | 74.05 | 74.75 | 47.5 | 90.23 | 74.28 | 7.53 | 79.11 |
| | CoT | 72.07 (-1.98) | 56.19 (-18.56) | 50.75 (+3.25) | 87.01 (-3.22) | 68.61 (-5.67) | 12.17 (+4.64) | 63.24 (-15.87) |
| | RCI | 66.25 (-7.80) | 66.61 (-8.14) | 55.21 (+7.71) | 94.36 (+4.13) | 70.96 (-3.32) | 14.58 (+7.05) | 57.99 (-21.12) |
| | CoVe | 73.25 (-0.80) | 58.36 (-16.39) | 57.95 (+10.45) | 91.36 (+1.13) | 85.59 (+11.31) | 36.22 (+28.69) | 85.29 (+6.18) |
| | Self-Refine | 75.46 (+1.41) | 73.79 (-0.96) | 58.29 (+10.79) | 88.19 (-2.04) | 78.34 (+4.06) | 32.77 (+25.24) | - |
| | Reflexion-v1 | 60.96 (-13.09) | 55.14 (-19.61) | 46.28 (-1.22) | 61.17 (-29.06) | 67.85 (-6.43) | 7.51 (-0.02) | - |
| Qwen2.5-72B | Base | 83.63 | 81.92 | 57.58 | 91.11 | 66.91 | 9.85 | 86.13 |
| | CoT | 89.87 (+6.24) | 88.44 (+6.52) | 57.34 (-0.24) | 94.14 (+3.03) | 63.71 (-3.20) | 16.19 (+6.34) | 62.45 (-23.68) |
| | RCI | 84.63 (+1.00) | 80.42 (-1.50) | 64.12 (+6.54) | 94.92 (+3.81) | 60.08 (-6.83) | 13.72 (+3.87) | 79.73 (-6.40) |
| | CoVe | 88.95 (+5.32) | 80.79 (-1.13) | 75.17 (+17.59) | 94.42 (+3.31) | 63.73 (-3.18) | 32.41 (+22.56) | 63.56 (-22.57) |
| | Self-Refine | 90.59 (+6.96) | 84.91 (+2.99) | 60.07 (+2.49) | 93.12 (+2.01) | 78.71 (+11.80) | 31.19 (+21.34) | - |
| | Reflexion-v1 | 69.65 (-13.98) | 66.54 (-15.38) | 40.95 (-16.63) | 71.84 (-19.27) | 56.28 (-10.63) | 11.23 (+1.38) | - |
| Claude3.5 | Base | 88.29 | 80.25 | 81.26 | 95.81 | 83.51 | 32.34 | 84.69 |
| | CoT | 92.3 (+4.01) | 82.48 (+2.23) | 64.51 (-16.75) | 97.55 (+1.74) | 85.48 (+1.97) | 16.67 (-15.67) | 80.53 (-4.16) |
| | RCI | 86.24 (-2.05) | 86.22 (+5.97) | 86.09 (+4.83) | 93.53 (-2.28) | 85.08 (+1.57) | 21.98 (-10.36) | 83.31 (-1.38) |
| | CoVe | 88.5 (+0.21) | 90.58 (+10.33) | 82.86 (+1.60) | 98.01 (+2.20) | 84.98 (+1.47) | 43.22 (+10.88) | 84.28 (-0.41) |
| | Self-Refine | 93.39 (+5.10) | 86.64 (+6.39) | 79.72 (-1.54) | 98.39 (+2.58) | 87.22 (+3.71) | 58.95 (+26.61) | - |
| | Reflexion-v1 | 81.32 (-6.97) | 68.08 (-12.17) | 69.6 (-11.66) | 77.71 (-18.10) | 81.1 (-2.41) | 42.68 (+10.34) | - |
| GPT-3.5 | Base | 82.94 | 77.92 | 55.15 | 79.14 | 70.44 | 27.29 | 80.29 |
| | CoT | 89.34 (+6.40) | 81.47 (+3.55) | 65.56 (+10.41) | 97.41 (+18.27) | 74.91 (+4.47) | 17.84 (-9.45) | 76.77 (-3.52) |
| | RCI | 87.17 (+4.23) | 86.92 (+9.00) | 79.52 (+24.37) | 93.46 (+14.32) | 69.78 (-0.66) | 23.31 (-4.12) | 78.14 (-2.15) |
| | CoVe | 85.17 (+2.23) | 89.67 (+11.75) | 81.49 (+26.34) | 94.98 (+15.84) | 83.75 (+13.31) | 48.5 (+21.21) | 83.95 (+3.66) |
| | Self-Refine | 91.47 (+8.53) | 82.64 (+4.72) | 69.8 (+14.64) | 98.18 (+19.04) | 85.34 (+14.9) | 52.14 (+24.85) | - |
| | Reflexion-v1 | 71.88 (-11.06) | 62.11 (-15.81) | 44.76 (-10.39) | 72.74 (-6.4) | 59.12 (-11.32) | 21.85 (-5.44) | - |
| GPT-4o | Base | 89.16 | 80.65 | 65.82 | 91.15 | 69.54 | 22.49 | 77.04 |
| | CoT | 91.86 (+2.70) | 81.68 (+1.03) | 61.45 (-4.37) | 97.81 (+6.66) | 73.46 (+3.92) | 13.75 (-8.74) | 64.58 (-12.46) |
| | RCI | 91.82 (+2.66) | 81.73 (+1.08) | 71.23 (+5.41) | 95.54 (+4.39) | 77.83 (+8.29) | 18.12 (-4.37) | 81.31 (+4.27) |
| | CoVe | 90.09 (+0.93) | 88.85 (+8.20) | 75.43 (+9.61) | 95.89 (+4.74) | 85.83 (+16.29) | 45.09 (+22.60) | 85.61 (+8.57) |
| | Self-Refine | 95.66 (+6.50) | 88.71 (+8.06) | 83.33 (+17.49) | 96.66 (+5.51) | 76.03 (+6.49) | 51.93 (+29.44) | - |
| | Reflexion-v1 | 71.51 (-17.65) | 64.62 (-16.03) | 48.32 (-17.50) | 79.56 (-11.59) | 56.58 (-12.96) | 15.71 (-6.78) | - |
| QWQ-32B | Base | 62.43 | 82.78 | 52.42 | 63.41 | 73.78 | 10.85 | 19.86 |
| | CoT | 65.86 (+3.43) | 75.23 (-7.55) | 62.43 (+10.01) | 81.41 (+18.00) | 75.62 (+1.84) | 11.31 (+0.46) | 14.19 (-5.67) |
| | RCI | 78.48 (+16.05) | 67.89 (-14.89) | 59.94 (+7.52) | 57.03 (-6.38) | 61.94 (-11.84) | 13.43 (+2.58) | 12.87 (-6.99) |
| | CoVe | 76.09 (+13.66) | 72.96 (-9.82) | 59.34 (+6.92) | 93.59 (+30.18) | 66.12 (-7.66) | 24.61 (+13.76) | 56.73 (+36.87) |
| | Self-Refine | 83.95 (+21.52) | 82.87 (+0.09) | 69.58 (+17.16) | 83.77 (+20.36) | 81.42 (+7.64) | 26.38 (+15.53) | - |
| | Reflexion-v1 | 54.66 (-7.77) | 62.86 (-19.92) | 42.83 (-9.59) | 52.97 (-10.44) | 64.25 (-9.53) | 21.54 (+10.69) | - |
| DeepSeek-V3 | Base | 89.29 | 83.35 | 74.79 | 95.12 | 85.02 | 31.35 | 91.67 |
| | CoT | 90.08 (+0.79) | 80.08 (-3.27) | 72.67 (-2.12) | 98.13 (+3.01) | 73.73 (-11.29) | 23.91 (-7.44) | 80.92 (-10.75) |
| | RCI | 84.62 (-4.67) | 73.72 (-9.63) | 69.85 (-4.94) | 95.36 (+0.24) | 86.13 (+1.11) | 27.23 (-4.12) | 89.34 (-2.33) |
| | CoVe | 88.72 (-0.57) | 83.34 (-0.01) | 80.64 (+5.85) | 94.79 (-0.33) | 72.71 (-12.31) | 52.17 (+20.82) | 82.57 (-9.10) |
| | Self-Refine | 92.34 (+3.05) | 82.78 (-0.57) | 85.97 (+11.18) | 99.15 (+4.03) | 84.87 (-0.15) | 51.13 (+19.78) | - |
| | Reflexion-v1 | 75.17 (-14.12) | 77.06 (-6.29) | 67.77 (-7.02) | 69.93 (-25.19) | 64.45 (-20.57) | 29.47 (-1.88) | - |

## H.2 All Performance Results for External Correction

Table 10: Performance of External Correction Methods on Different LLMs

| LLM | Method | HotpotQA(↑) | CS-QA(↑) | AQUA(↑) | GSM8K(↑) | MATH(↑) | GPQA(↑) | HumanEval(↑) |
|---|---|---|---|---|---|---|---|---|
| LLaMA3.1-8B | Base | 75.8 | 76.16 | 53.88 | 81.55 | 74.37 | 9.74 | 73.44 |
| | CoT | 72.14 (-3.66) | 73.85 (-2.31) | 44.55 (-9.33) | 78.28 (-3.27) | 77.65 (+3.28) | 13.11 (+3.37) | 51.83 (-21.61) |
| | Reflexion-v2 | 89.02 (+13.22) | 78.66 (+2.5) | 59.77 (+5.89) | 83.9 (+2.35) | 78.14 (+3.77) | 22.14 (+12.4) | - |
| | RARR | 81.31 (+5.51) | 74.6 (-1.56) | 56.54 (+2.66) | 79.31 (-2.24) | 84.43 (+10.06) | 34.58 (+24.84) | 67.58 (-5.86) |
| | RATT | 76.82 (+1.02) | 76.72 (+0.56) | 54.94 (+1.06) | 72.34 (-9.21) | 81.16 (+6.79) | 13.83 (+4.09) | 62.33 (-11.11) |
| | CRITIC | 69.33 (-6.47) | - | - | 69.12 (-12.43) | - | - | - |
| LLaMA3.1-70B | Base | 81.28 | 81.88 | 62.65 | 90.63 | 78.21 | 15.62 | 62.18 |
| | CoT | 86.13 (+4.85) | 82.84 (+0.96) | 62.93 (+0.28) | 95.92 (+5.29) | 60.11 (-18.1) | 23.74 (+8.12) | 46.39 (-15.79) |
| | Reflexion-v2 | 88.78 (+7.5) | 86.17 (+4.29) | 62.45 (-0.2) | 82.11 (-8.52) | 84.82 (+6.61) | 24.62 (+9) | - |
| | RARR | 90.49 (+9.21) | 79.93 (-1.95) | 67.29 (+4.64) | 88.5 (-2.13) | 87.8 (+9.59) | 35.79 (+20.17) | 78.96 (+16.78) |
| | RATT | 82.37 (+1.09) | 82.08 (+0.2) | 68.22 (+5.57) | 86.15 (-4.48) | 85.79 (+7.58) | 26.09 (+10.47) | 66.05 (+3.87) |
| | CRITIC | 85.57 (+4.29) | - | - | 95.24 (+4.61) | - | - | - |
| Qwen2.5-7B | Base | 74.05 | 74.75 | 47.5 | 90.23 | 74.28 | 7.53 | 79.11 |
| | CoT | 72.07 (-1.98) | 56.19 (-18.56) | 50.75 (+3.25) | 87.01 (-3.22) | 68.61 (-5.67) | 12.17 (+4.64) | 63.24 (-15.87) |
| | Reflexion-v2 | 87.21 (+13.16) | 67.95 (-6.8) | 54.35 (+6.85) | 89.31 (-0.92) | 85.63 (+11.35) | 26.18 (+18.65) | - |
| | RARR | 71.67 (-2.38) | 75.52 (+0.77) | 45.61 (-1.89) | 85.22 (-5.01) | 79.44 (+5.16) | 34.44 (+26.91) | 85.71 (+6.6) |
| | RATT | 72.4 (-1.65) | 84.55 (+9.8) | 65.17 (+17.67) | 94.92 (+4.69) | 84.61 (+10.33) | 14.61 (+7.08) | 80.69 (+1.58) |
| | CRITIC | 69.89 (-4.16) | - | - | 74.42 (-15.81) | - | - | - |
| Qwen2.5-72B | Base | 83.63 | 81.92 | 57.58 | 91.11 | 66.91 | 9.85 | 86.13 |
| | CoT | 89.87 (+6.24) | 88.44 (+6.52) | 57.34 (-0.24) | 94.14 (+3.03) | 63.71 (-3.2) | 16.19 (+6.34) | 62.45 (-23.68) |
| | Reflection-v2 | 83.69 (+0.06) | 82.33 (+0.41) | 57.46 (-0.12) | 92.64 (+1.53) | 77.46 (+10.55) | 14.72 (+4.87) | - |
| | RARR | 87.77 (+4.14) | 86.16 (+4.24) | 57.23 (-0.35) | 86.4 (-4.71) | 73.66 (+6.75) | 40.13 (+30.28) | 87.63 (+1.5) |
| | RATT | 84.68 (+1.05) | 84.12 (+2.2) | 64.41 (+6.83) | 90.64 (-0.47) | 77.94 (+11.03) | 25.74 (+15.89) | 84.72 (-1.41) |
| | CRITIC | 84.78 (+1.15) | - | - | 79.35 (-11.76) | - | - | - |
| Claude3.5 | Base | 88.29 | 80.25 | 81.26 | 95.81 | 83.51 | 32.34 | 84.69 |
| | CoT | 92.3 (+4.01) | 82.48 (+2.23) | 64.51 (-16.75) | 97.55 (+1.74) | 85.48 (+1.97) | 16.67 (-15.67) | 80.53 (-4.16) |
| | Reflexion-v2 | 93.62 (+5.33) | 91.77 (+11.52) | 84.83 (+3.57) | 94.1 (-1.71) | 92.56 (+9.05) | 53.21 (+20.87) | - |
| | RARR | 87.44 (-0.85) | 80.79 (+0.54) | 83.52 (+2.26) | 94.36 (-1.45) | 94.18 (+10.67) | 50.34 (+18.0) | 86.27 (+1.58) |
| | RATT | 82.13 (-6.16) | 77.62 (-2.63) | 88.51 (+7.25) | 96.39 (+0.58) | 97.31 (+13.8) | 38.28 (+5.94) | 85.29 (+0.6) |
| | CRITIC | 95.16 (+6.87) | - | - | 94.85 (-0.96) | - | - | - |
| GPT-3.5 | Base | 82.94 | 77.92 | 55.15 | 79.14 | 70.44 | 27.29 | 80.29 |
| | CoT | 89.34 (+6.4) | 81.47 (+3.55) | 65.56 (+10.41) | 97.41 (+18.27) | 74.91 (+4.47) | 17.84 (-9.45) | 76.77 (-3.52) |
| | Reflexion-v2 | 90.31 (+7.37) | 82.72 (+4.8) | 76.22 (+21.07) | 92.62 (+13.48) | 79.84 (+9.4) | 30.51 (+3.22) | - |
| | RARR | 82.37 (-0.57) | 74.04 (-3.88) | 62.04 (+6.89) | 89.93 (+10.79) | 73.36 (+2.92) | 28.42 (+1.13) | 78.58 (-1.71) |
| | RATT | 75.92 (-7.02) | 78.39 (+0.47) | 64.99 (+9.84) | 88.27 (+9.13) | 73.66 (+3.22) | 37.87 (+10.58) | 80.56 (+0.27) |
| | CRITIC | 82.49 (-0.45) | - | - | 82.72 (+3.58) | - | - | - |
| GPT-4o | Base | 89.16 | 80.65 | 65.82 | 91.15 | 69.54 | 22.49 | 77.04 |
| | CoT | 91.86 (+2.7) | 81.68 (+1.03) | 61.45 (-4.37) | 97.81 (+6.66) | 73.46 (+3.92) | 13.75 (-8.74) | 64.58 (-12.46) |
| | Reflexion-v2 | 91.17 (+2.01) | 84.23 (+3.58) | 77.01 (+11.19) | 94.13 (+2.98) | 73.46 (+3.92) | 19.47 (-3.02) | - |
| | RARR | 90.89 (+1.73) | 84.87 (+4.22) | 82.13 (+16.31) | 95.34 (+4.19) | 81.61 (+12.07) | 35.69 (+13.2) | 94.09 (+17.05) |
| | RATT | 90.84 (+1.68) | 80.49 (-0.16) | 77.23 (+11.41) | 98.73 (+7.58) | 73.48 (+3.94) | 24.95 (+2.46) | 86.04 (+9.0) |
| | CRITIC | 91.08 (+1.92) | - | - | 97.44 (+6.29) | - | - | - |
| QWQ-32B | Base | 62.43 | 82.78 | 52.42 | 63.41 | 73.78 | 10.85 | 19.86 |
| | CoT | 65.86 (+3.43) | 75.23 (-7.55) | 62.43 (+10.01) | 81.41 (+18.0) | 75.62 (+1.84) | 11.31 (+0.46) | 14.19 (-5.67) |
| | Reflexion-v2 | 83.18 (+20.75) | 81.71 (-1.07) | 73.55 (+21.13) | 83.75 (+20.34) | 73.56 (-0.22) | 23.12 (+12.27) | - |
| | RARR | 88.28 (+25.85) | 85.03 (+2.25) | 72.67 (+20.25) | 86.33 (+22.92) | 82.97 (+9.19) | 31.66 (+20.81) | 34.92 (+15.06) |
| | RATT | 66.19 (+3.76) | 76.78 (-6.0) | 50.88 (-1.54) | 72.85 (+9.44) | 66.12 (-7.66) | 15.62 (+4.77) | 22.71 (+2.85) |
| | CRITIC | 79.22 (+16.79) | - | - | 90.83 (+27.42) | - | - | - |
| DeepSeek-V3 | Base | 89.29 | 83.35 | 74.79 | 95.12 | 85.02 | 31.35 | 91.67 |
| | CoT | 90.08 (+0.79) | 80.08 (-3.27) | 72.67 (-2.12) | 98.13 (+3.01) | 73.73 (-11.29) | 23.91 (-7.44) | 80.92 (-10.75) |
| | Reflexion-v2 | 84.84 (-4.45) | 84.38 (+1.03) | 68.47 (-6.32) | 96.28 (+1.16) | 82.62 (-2.4) | 27.72 (-3.63) | - |
| | RARR | 88.99 (-0.3) | 84.15 (+0.8) | 74.28 (-0.51) | 94.93 (-0.19) | 87.55 (+2.53) | 40.31 (+8.96) | 82.41 (-9.26) |
| | RATT | 84.93 (-4.36) | 86.53 (+3.18) | 78.19 (+3.4) | 92.47 (-2.65) | 86.57 (+1.55) | 36.14 (+4.79) | 92.55 (+0.88) |
| | CRITIC | 91.72 (+2.43) | - | - | 94.37 (-0.75) | - | - | - |

