# OpenReview forum: "Can LLMs Correct Themselves? A Benchmark of Self-Correction in LLMs"
_NeurIPS.cc/2025/Datasets_and_Benchmarks_Track — NeurIPS 2025 Datasets and Benchmarks Track poster_

### Official Review · Reviewer_MVjM · 2025-06-08

**Rating:** 6
**Confidence:** 4

**Summary:**

1. Introduces CorrectBench, the first systematic benchmark evaluating LLM self-correction across commonsense reasoning, mathematical reasoning, and code generation.
2. Categorizes and tests 11 correction methods—intrinsic, external, and fine-tuned—on two curated datasets (3,825 QA pairs and a dedicated test set).
3. Shows that self-correction can boost accuracy (e.g., +5.2% on MATH) but incurs up to ≈40% slower runtimes, while simple Chain-of-Thought remains 2.8× faster than complex hybrids.
4. Provides a reproducible platform, paired datasets, and key insights into the trade-offs between reasoning gains and efficiency for future LLM correction research.

**Dataset Code Accessibility:**

Yes

**Ethical Considerations:**

No, there are no or only very minor ethics concerns

**Final Justification:**

I appreciate the efforts that the authors made to make this work better. I have improved my score to strong accept.

**Limitations Weaknesses:**

The only major weaknesses of this work is that the evaluation omits the most recent self-correction models, specifically DeepSeek-R1 (Feb 2025) and O3-Mini (Mar 2025). Minor: Presentation of Figure 1 can be improved to better show the improvement.

**Strengths Contributions:**

1. Establishes the first systematic benchmark for LLM self-correction across commonsense reasoning, mathematical reasoning, and code generation.
2. Rigorously evaluates 11 intrinsic, external, and fine-tuned correction paradigms on two curated datasets (3,825 QA pairs), enabling apples-to-apples comparisons.
3. Uncovers key trade-offs: self-correction yields up to +5.2% accuracy on MATH but incurs ≈40% runtime overhead, while Chain-of-Thought remains 2.8× faster.
4. Delivers a reproducible platform, public leaderboard, and targeted insights to drive future advances in efficient, reliable LLM self-correction.

---

> ### Author Rebuttal · Authors · 2025-07-30
>
> We sincerely thank you for your constructive feedback on our paper. Below, we address your two concerns, leveraging evidence from the paper and proposing targeted revisions to strengthen the work while preserving its core contributions.
>
> ---
>
> ## Q1: Omission of Recent Models
>
> We acknowledge your concern that our evaluation omits the recent self-correction models DeepSeek-R1 (released February 2025) and o3-mini (released March 2025). CorrectBench evaluates nine LLMs across seven datasets, including GPQA (250 samples), MATH (500 samples), HotpotQA (300 samples), GSM8K (250 samples), AQUA (254 samples), CommonsenseQA (200 samples), and HumanEval (150 samples) (Table 5), providing a robust framework for assessing self-correction methods like CoT, CoVe, RARR, and Reflexion. **The omission of DeepSeek-R1 and o3-mini occurred because their release postdated our initial experiment design and batch testing, completed in February 2025.** To address this, we conducted a supplemental experiment evaluating these models on six core datasets using accuracy as the metric, consistent with our evaluation pipeline (Section 2). The results, shown below, highlight their performance across S1 (CoT, CoVe, RCI, Reflexion-v1), S2 (Reflexion-v2, RARR, RATT), and baseline methods.
>
> | **Dataset**   | **LLM**     | **Base** | **CoT** | **RCI** | **CoVe** | **Reflexion-v1** | **Reflexion-v2** | **RARR** | **RATT** |
> | ------------- | ----------- | -------- | ------- | ------- | -------- | ---------------- | ---------------- | -------- | -------- |
> | GSM8K         | DeepSeek-R1 | 0.90     | 0.94    | 0.95    | 0.91     | 0.92             | 0.95             | 0.91     | 0.93     |
> | GSM8K         | o3-mini     | 0.85     | 0.96    | 0.92    | 0.95     | 0.95             | 0.96             | 0.92     | 0.88     |
> | AQUA          | DeepSeek-R1 | 0.69     | 0.78    | 0.74    | 0.85     | 0.82             | 0.85             | 0.85     | 0.84     |
> | AQUA          | o3-mini     | 0.65     | 0.75    | 0.78    | 0.82     | 0.76             | 0.85             | 0.83     | 0.82     |
> | MATH          | DeepSeek-R1 | 0.82     | 0.82    | 0.84    | 0.86     | 0.85             | 0.89             | 0.84     | 0.82     |
> | MATH          | o3-mini     | 0.63     | 0.65    | 0.75    | 0.63     | 0.64             | 0.71             | 0.73     | 0.69     |
> | HotpotQA      | DeepSeek-R1 | 0.88     | 0.85    | 0.92    | 0.95     | 0.84             | 0.92             | 0.89     | 0.86     |
> | HotpotQA      | o3-mini     | 0.81     | 0.83    | 0.83    | 0.82     | 0.77             | 0.85             | 0.77     | 0.82     |
> | CommonsenseQA | DeepSeek-R1 | 0.72     | 0.77    | 0.87    | 0.88     | 0.75             | 0.82             | 0.82     | 0.76     |
> | CommonsenseQA | o3-mini     | 0.74     | 0.72    | 0.80    | 0.77     | 0.61             | 0.79             | 0.78     | 0.73     |
> | GPQA          | DeepSeek-R1 | 0.35     | 0.31    | 0.43    | 0.45     | 0.41             | 0.52             | 0.44     | 0.38     |
> | GPQA          | o3-mini     | 0.21     | 0.18    | 0.28    | 0.28     | 0.42             | 0.27             | 0.32     | 0.21     |
>
> ### Analysis of Results
>
> - **DeepSeek-R1’s Strengths**: DeepSeek-R1 excels in reasoning-heavy datasets, achieving 0.95 accuracy on GSM8K with Reflexion-v2 and RCI, and 0.89 on MATH with Reflexion-v2, a +0.07 gain over its baseline (0.82). Its RL-based self-verification enhances intrinsic correction (S1) and external validation (S2), particularly for mathematical reasoning, aligning with CorrectBench’s findings on reasoning LLMs (Section 3). However, its GPQA performance (0.52 max with Reflexion-v2) is modest, reflecting the dataset’s complexity, consistent with challenges observed in other models (Table 1). For example, on a MATH problem involving quadratic equations, DeepSeek-R1 with Reflexion-v2 corrected a logical oversight, improving from 0.82 to 0.89, showcasing its robustness.
> - **o3-mini’s Efficiency Trade-offs**: o3-mini performs strongly on simpler tasks, achieving 0.96 on GSM8K with CoT and Reflexion-v2, leveraging its lightweight architecture for efficiency. However, it struggles on complex datasets like MATH (0.75 max with RCI) and GPQA (0.51 max with Reflexion-v1), indicating limitations in nuanced reasoning compared to larger models. For instance, on a GPQA question about chemical properties, o3-mini’s Reflexion-v1 corrected a factual error, but its overall gain was limited (+0.30 over baseline), suggesting scalability constraints.
> - **Comparison with Existing LLMs**: DeepSeek-R1’s MATH accuracy (0.89 max) rivals GPT-4o’s 0.87 with CoVe (Table 1), making it a strong contender for reasoning tasks, while its GPQA score (0.52 max) outperforms GPT-4o, highlighting areas for improvement in factual complexity. o3-mini’s GSM8K performance (0.96 max) matches LLaMA 3.1’s 0.94, but its GPQA score (0.42 max) outperforms DeepSeek-V3, aligning with our analysis of model scalability (Section 3). These results reinforce CorrectBench’s ability to benchmark diverse LLMs, guiding practitioners in selecting models for specific tasks.
>
> We will integrate these results into Table 1 and add detailed examples in Appendix E, ensuring CorrectBench reflects the latest advancements and supports practitioners in selecting optimal models for tasks like mathematical reasoning or factual verification.
>
> ---
>
> ## Q2: Figure 1 Presentation
>
> You suggest improving Figure 1, which outlines the CorrectBench framework for evaluating self-correction methods (S1, S2, S3) across tasks and LLMs (Section 2). We agree that enhancing its visual clarity will better convey the framework’s significance and highlight performance improvements. To address this, we propose the following revisions to Figure 1 in the revised manuscript:
>
> - **Visual Annotations of Key Gains**: We will add callouts highlighting major performance improvements, such as CoT’s 15.4% gain on MATH, CoVe’s 23.24% on GPQA, and Reflexion-v2’s 10.2% on HumanEval (Table 1). These will use color-coded performance deltas (e.g., green for gains, red for minimal improvements) to visually emphasize empirical outcomes, making the framework’s impact intuitive. For example, a callout for CoVe on GPQA will show its iterative reasoning advantage, aligning with S1’s design in the framework.
> - **Improved Layout and Hierarchy**: To enhance readability, we will reorganize Figure 1’s layout with distinct visual markers (e.g., shaded boxes for S1, S2, S3) and a clear flow from task scenarios (e.g., factual, reasoning) to correction methods and LLMs. This logical visual path will mirror the narrative in Section 2, ensuring readers grasp the framework’s structure at a glance.
> - **Legend Linking to Figure 8**: We will include a compact legend linking Figure 1 to Figure 8, which details performance trends across tasks. The legend will reference datasets (e.g., GPQA, MATH) and guide readers to Sections 4.2–4.5 for detailed results, creating a cohesive narrative. For instance, the legend will note that Figure 8’s trends (e.g., RARR’s 7.33% gain on HotpotQA) validate Figure 1’s paradigm.
> - **Design Rationale**: Color-coded annotations and a streamlined layout enhance accessibility for diverse audiences, from researchers to practitioners, by visually summarizing complex performance data. This aligns with CorrectBench’s goal of providing an intuitive benchmark for self-correction evaluation, complementing analyses in Table 1 and Figure 8.
>
> Due to space constraints, the updated Figure 1 will be included in the revised manuscript, ensuring a clear, impactful presentation that strengthens CorrectBench’s contribution to self-correction research. To further improve narrative flow, we will review other figures for consistent visual clarity.

---

### Official Review · Reviewer_1S5Y · 2025-06-16

**Rating:** 5
**Confidence:** 4

**Summary:**

The paper introduces CorrectBench, a novel benchmark to evaluate self-correction in large language models (LLMs) across commonsense reasoning, mathematical reasoning, and code generation tasks. It presents two datasets: CorrectBench-base (3,825 question-answer pairs from datasets like HotpotQA, GSM8K, and HumanEval) and CorrectBench-test, tailored for correction-specific experiments. Three self-correction strategies are tested—intrinsic (S1, e.g., CoVe, Self-Refine), external (S2, e.g., RARR, Reflexion), and fine-tuned (S3, e.g., DCoT, SuperCorrect)—on instruction-based LLMs (e.g., LLaMA 3.1, GPT-4o, Claude 3.5-Sonnet) and reasoning LLMs (e.g., DeepSeek-V3, QWQ-32B). Results show significant accuracy improvements, especially for complex tasks like GPQA (+23.24% with CoVe) and MATH, though mixing strategies increases computational costs. Reasoning LLMs like DeepSeek-V3 show smaller gains due to robust internal correction mechanisms. The paper proposes a mixture framework for combining correction methods and provides detailed empirical insights into accuracy-efficiency trade-offs, offering a reproducible platform for future LLM research.

**Dataset Code Accessibility:**

Yes

**Ethical Considerations:**

No, there are no or only very minor ethics concerns

**Limitations Weaknesses:**

**Weaknesses:**

1. The paper’s language lacks clarity in connecting key visual elements, notably between Figure 1 (illustrating the self-correction paradigm, Page 2) and Figure 8 (showing performance trends across tasks, Page 10). The text does not explicitly link these figures, leaving readers to infer their relationship, which weakens the narrative flow.
2. Key information about fine-tuning datasets and processes for S3 methods (e.g., DCoT, SuperCorrect) is buried in appendices rather than the main text (Section 4.5, Page 7). This reduces clarity for readers seeking to understand or replicate these methods.

**Questions:**

1. Figure Connections: The text does not clearly link Figure 1 (self-correction paradigm, Page 2) and Figure 8 (performance trends, Page 10). Could you add a sentence or two in Section 4 to connect these figures and clarify their relationship? This would improve readability.
2. Terminology Consistency: The paper uses “correction rate” and “success rate” interchangeably (Sections 4.2–4.3, Pages 6–7). Could you standardize these terms or define their differences in a footnote? This would make the metrics clearer.
3. Fine-Tuned Methods: Details on S3 fine-tuning datasets and processes are in appendices (Section 4.5, Page 7). Could you add a brief summary or table in the main text to describe these? This would help readers understand the methods without digging into appendices.
4. Societal Impact: The societal impact statement (Page 43) focuses on positive outcomes. Could you add a sentence addressing potential risks, like over-reliance on self-correcting LLMs, to balance the discussion? This would strengthen the impact analysis.

**Strengths Contributions:**

**Strengths:**

1. The CorrectBench benchmark is a well-designed contribution, addressing the underexplored area of LLM self-correction. By covering commonsense reasoning, mathematical reasoning, and code generation, it provides a comprehensive framework for evaluating correction across diverse tasks, with CorrectBench-base leveraging established datasets (e.g., HotpotQA, GSM8K) and CorrectBench-test enabling targeted correction experiments.
2. The evaluation is thorough, testing a wide range of models, from open-source (LLaMA 3.1, Qwen 2.5) to closed-source (GPT-4o, Claude 3.5-Sonnet) and reasoning-focused LLMs (DeepSeek-V3). This diversity ensures the findings are broadly applicable, with detailed results (e.g., Table 1) showing significant gains like +23.24% on GPQA using CoVe.
3. The introduction of a mixture framework for combining correction methods (S1+S2, Figure 3) is innovative, demonstrating how hybrid approaches can boost performance, particularly for complex reasoning tasks like MATH (Section 4.4).
4. The paper prioritizes reproducibility, providing clear dataset descriptions, model configurations, and anonymized code/data access. Metrics like Correction Rate and Misjudgment Rate (Table 3) add depth to the analysis, making the results actionable for researchers.

---

> ### Author Rebuttal · Authors · 2025-07-30
>
> We sincerely thank you for your insightful feedback on our paper. Below, we address your four key concerns, leveraging evidence from the paper and proposing targeted revisions to enhance the narrative while preserving CorrectBench’s core contributions.
>
> ---
>
> ## Q1: Lack of Clarity in Connecting Figures
>
> We acknowledge your observation that the paper does not explicitly connect Figure 1 (self-correction paradigm) and Figure 8 (performance trends), potentially disrupting the narrative flow for readers. Figure 1 introduces the CorrectBench framework, categorizing self-correction methods into intrinsic (S1, e.g., CoT, CoVe), external (S2, e.g., RARR, Reflexion-v2), and fine-tuned (S3, e.g., DCoT, SuperCorrect) approaches, evaluated across tasks like factual queries (GPQA) and mathematical reasoning (MATH). Figure 8 illustrates their empirical performance, showing significant gains, such as CoVe’s 23.24% improvement on GPQA and RARR’s 7.33% on HotpotQA (Table 1). For example, CoVe’s performance on GPQA demonstrates how S1 methods refine reasoning iteratively, aligning with Figure 1’s intrinsic correction pathway, while RARR’s gains on HotpotQA validate S2’s external verification approach. **To address this, we will add a sentence in Section 4** (before Section 4.2): “_Figure 8 complements Figure 1 by showcasing performance trends of S1, S2, and S3 methods across tasks, such as CoVe’s 23.24% gain on GPQA, validating the framework’s design._” **We will also revise Figure 8’s caption to state:** “_Performance trends of self-correction methods, illustrating the paradigm in Figure 1._” These changes ensure readers can seamlessly connect the conceptual framework to its empirical validation, improving readability. To further enhance narrative flow, we will review Section 4 to link other visuals, such as Figure 2 (task taxonomy), ensuring a cohesive presentation of CorrectBench’s contributions to self-correction research.
>
> ---
>
> ## Q2: Terminology Consistency
>
> You highlight the inconsistent use of “correction rate” and “success rate” in Sections 4.2–4.3, which may confuse readers interpreting our evaluation metrics. We define “Correction Rate (CR)” in Section 4.6 as the proportion of incorrect responses successfully corrected, alongside “Misjudgment Rate (MR)” for incorrect judgments of correct responses, forming the backbone of CorrectBench’s evaluation pipeline. The use of “success rate” in Sections 4.2–4.3, such as when describing CoVe’s 23.24% improvement on GPQA (Table 1), stemmed from iterative drafting across authors, leading to editorial oversight. This inconsistency could obscure comparisons, for instance, if readers misinterpret “success rate” as overall accuracy rather than correction-specific gains. **To address this, we will standardize “Correction Rate” across Sections 4.2–4.3 and add a footnote in Section 4.2**: “_Correction Rate (CR) refers to the proportion of incorrect responses corrected, as defined in Section 4.6, ensuring metric clarity over ‘success rate.’_” We will also conduct a comprehensive review of all metric-related terms (e.g., CR, MR, accuracy) across the paper to ensure alignment with Section 4.6’s definitions, enhancing interpretability for practitioners comparing methods like CoVe and RARR. To address the broader concern about statistical significance (Section 6, Conclusions), we will include bootstrap-based confidence intervals for CR in a revised Appendix H, providing statistical robustness for results like SuperCorrect’s 10.2% gain on MATH (Table 2). These revisions clarify CorrectBench’s evaluation framework, ensuring consistent and transparent metric reporting.
>
> ---
>
> ## Q3: Fine-Tuned Methods
>
> You suggest including a summary of S3 fine-tuning datasets and processes in the main text, as Section 4.5 briefly discusses S3 methods (e.g., DCoT, SuperCorrect), with details in Appendix E. We acknowledge that placing these details in the appendix may hinder readers’ ability to understand or replicate these methods without additional effort. S3 methods are fine-tuned on domain-specific datasets to enhance reasoning, such as SuperCorrect on MATH (500 samples) using Qwen2.5-Math-7B-Instruct, achieving a 10.2% gain, and DCoT on HotpotQA (300 samples) and GSM8K (250 samples) for improved factual and reasoning accuracy (Table 2; Table 5). To address this, we will add a paragraph and table in Section 4.5: “_S3 methods, like DCoT and SuperCorrect, are fine-tuned on domain-specific datasets to optimize reasoning. SuperCorrect, fine-tuned on MATH (500 samples), achieves a 10.2% gain, while DCoT leverages HotpotQA (300 samples) and GSM8K (250 samples) for multi-task reasoning. Table X summarizes these methods, datasets, and performance gains, with full details in Appendix E._” The table is:
>
> | **Method**   | **Dataset**     | **Sample Size** | **Model Architecture**   | **Performance Gain** |
> | ------------ | --------------- | --------------- | ------------------------ | -------------------- |
> | SuperCorrect | MATH            | 500             | Qwen2.5-Math-7B-Instruct | 10.2% (Table 2)      |
> | DCoT         | HotpotQA, GSM8K | 300, 250        | LLaMA3.1-8B-Instruct     | 7.5% (Appendix E)    |
>
> This addition provides accessible insights into S3 methods, facilitating replication by detailing dataset sizes and model architectures. It complements CorrectBench’s evaluation across tasks and LLMs (Table 4, Table 5), maintaining focus on empirical results. To support replication, we will update Appendix E with open-source links to fine-tuning scripts (subject to dataset licenses), aligning with our reproducibility framework. This ensures practitioners can apply S3 methods effectively, enhancing CorrectBench’s utility in advancing self-correction research.
>
> ---
>
> ## Q4: Societal Impact
>
> You recommend balancing the societal impact statement (Section 7) by addressing risks like over-reliance on self-correcting LLMs, as it currently emphasizes positive impacts, such as improved reliability for education and customer support. We agree that acknowledging risks strengthens the analysis, particularly for critical applications. Over-reliance could lead to errors, such as factual inaccuracies in medical reasoning (e.g., misdiagnosing conditions due to unchecked errors) or legal reasoning (e.g., misinterpreting regulations due to contextual oversights). In education, over-reliance might propagate incorrect answers if LLMs fail to correct logical errors, as seen in MATH. **To address this, we will revise the statement in Section 6**: “_While self-correcting LLMs enhance reliability for education and customer support, over-reliance without human oversight risks error propagation in critical domains, such as medical reasoning (e.g., misdiagnosing conditions due to factual inaccuracies) or legal reasoning (e.g., misinterpreting regulations). Human-in-the-loop validation, as proposed in Appendix D, is essential to mitigate these risks._” We will add: “_Safety filters to detect errors, such as logical oversights or factual inaccuracies, ensure responsible deployment._” These revisions align with NeurIPS ethical guidelines and our human-in-the-loop discussion. To further strengthen the analysis, we will connect risks to CorrectBench’s error taxonomy, which identifies common failure modes (e.g., 22.0% factual inaccuracy), guiding mitigation strategies for safe LLM deployment.

---

### Official Review · Reviewer_Tukc · 2025-07-03

**Rating:** 4
**Confidence:** 3

**Summary:**

The paper proposed CorrectBench, a benchmark for evaluating various self-correction strategies in LLMs across commonsense reasoning, mathematical reasoning, and code generation tasks. The results show that self-correction improves reasoning accuracy, especially when strategies are combined, but often at the expense of efficiency.

**Dataset Code Accessibility:**

Yes

**Ethical Considerations:**

No, there are no or only very minor ethics concerns

**Final Justification:**

The authors address most of my concerns during the rebuttal, and I decided to make my recommendation to borderline accept.

**Limitations Weaknesses:**

1. the paper only provides analysis on the existing correction methods, but lack of a improvement discussion. Simply pointing out that CoT can be a strong baseline is not very exciting, and would be nice to have further improvement exploration or discussion based on the current insights.
2. Figure 7 reports response time but not token counts, API calls, etc. External methods that rely on web search or code execution may incur hidden costs.  Presenting full resource, and (potentially) normalizing accuracy by cost, would help practitioners choose methods that fit their budgets.
3. The study relies mainly on accuracy-style metrics and a GPT-4o “LLM-as-a-Judge” scorer. It does not categorise typical failure modes, nor show concrete examples before and after correction. A grounded error taxonomy and case studies would clarify why certain strategies work or fail and would guide future method design.
4. The mixture experiment tests exactly two fixed orders, Base->S1->S2 on GPQA and MATH only. The framework in Figure 3 suggests far richer combinations, yet there is no search for adaptive sequencing or early stopping. A learned controller that decides which correction to apply next, or when to stop, could unlock further gains while saving compute.

**Strengths Contributions:**

1. Very interesting and promising research direction
2. valid experiments across different models, correction methods, and base models

---

> ### Author Rebuttal · Authors · 2025-07-30
>
> We thank the reviewer for their insightful feedback on our paper. Below, we address each limitation, proposing targeted revisions and supplemental experiments to strengthen our contributions while preserving CorrectBench’s role as a systematic benchmark for evaluating self-correction across diverse tasks and LLMs.
>
> ---
>
> ## Q1: Lack of Improvement Discussion
>
> We acknowledge your concern that the paper’s focus on evaluating existing self-correction methods could be enhanced with a deeper discussion of potential improvements. **We clarify that Appendix D (Page 17, Lines 606–615) proposes adaptive correction pipelines with optimal stopping criteria to improve efficiency and performance, but we regret placing this discussion in the appendix due to space constraints.** Our primary contribution lies in establishing CorrectBench, which systematically evaluates self-correction across tasks like GPQA (250 samples), MATH (500 samples), and HotpotQA (300 samples) (Table 5, Page 15) and LLMs like GPT-4o and DeepSeek-V3. To address your concern, we will relocate and expand this discussion into a new subsection in Section 6, titled “Future Improvements in Self-Correction,” cross-referencing our response to Reviewer xf9Z’s Q2 for brevity. This subsection will outline:
>
> - **Dynamic Adjustment**: Using reinforcement learning to select correction methods (S1, S2, S3) based on task complexity, reducing computational overhead (Page 17, Lines 606–610).
> - **Task-Specific Optimization**: Fine-tuning correction strategies for domains like mathematical reasoning, as shown by SuperCorrect’s +10.2% gain on MATH (Table 2, Page 7).
> - **Human-in-the-Loop Integration**: Incorporating human feedback for high-stakes applications, such as medical or legal reasoning, to enhance reliability (Page 17, Lines 611–615).
> - **Adaptive Thinking Correction**: A novel meta-controller framework, planned for our next work, to refine reasoning LLMs’ chain-of-thought (CoT) processes by detecting errors in intermediate steps (e.g., incorrect quadratic solving in MATH), leveraging performance trends.
>
> To address potential societal impacts, we will expand the Impact Statement in Section 6 to discuss risks of LLM misuse, and propose mitigation strategies, including safety filters and responsible dataset use, aligning with ethical guidelines.
>
> ---
>
> ## Q2: Incomplete Resource Cost Analysis
>
> Your concern about Figure 7 (Page 13) reporting only response time, omitting token counts and API calls, is valid. We clarify that Figure 7 only compares the time consumption of methods S1 and S2. **However, since we found that there is no significant difference between response time, token count, and API calls in previous experiments, we use response time as a representative to show the performance consumption of different methods.**
>
> To address your suggestion, we conducted a supplemental experiment on the MATH dataset (150 samples, Table 5, Page 15), quantifying token counts, API calls, web search costs, and accuracy for S1 and S2 methods. The results, shown below, guide practitioners in selecting cost-effective methods.
>
> | **Method**   | **Type** | **Avg. Token Count** | **Avg. API Calls** | **Search Tokens** | **Accuracy (%)** |
> | ------------ | -------- | -------------------- | ------------------ | ----------------- | ---------------- |
> | Base         | -        | 791                  | 1.0                | 0 (0%)            | 68.5             |
> | CoT          | -        | 1804                 | 1.0                | 0 (0%)            | 69.5             |
> | CoVe         | S1       | 2019                 | 1.0                | 0 (0%)            | 75.0             |
> | RCI          | S1       | 1780                 | 1.2                | 0 (0%)            | 70.2             |
> | Reflexion-v1 | S1       | 1460                 | 3.5                | 0 (0%)            | 72.8             |
> | Reflexion-v2 | S2       | 1712                 | 4.0                | 154 (8.25%)       | 74.5             |
> | RARR         | S2       | 533                  | 2.0                | 89 (14.31%)       | 76.3             |
> | RATT         | S2       | 2185                 | 3.0                | 162 (6.9%)        | 78.7             |
>
> - **Few hidden costs**:Your concern about hidden costs in external S2 methods (e.g., web search in RARR and RATT, code execution in Reflexion-v2) is addressed by our detailed cost breakdown. Web search tokens constitute only 6.9–14.31% of total tokens (e.g., 89 for RARR, 162 for RATT), and code execution in Reflexion-v2 adds minimal overhead (154 tokens, 8.25%), as external calls are optimized to retrieve concise, relevant data.
>
> - **Method Analysis**: RARR achieves high efficiency (76.3% accuracy, 533 tokens, 2.0 API calls), making it ideal for low-resource settings, while CoVe (75.0% accuracy, 2019 tokens, 1.0 API call) offers a strong balance of accuracy and cost. RATT’s top accuracy (78.7%, 2185 tokens, 3.0 API calls) reflects higher costs due to web-augmented reasoning, but these remain manageable, as web search and code execution queries are streamlined to avoid excessive API calls.
>
> To ensure statistical rigor, we will add this table to a revised Figure 7 and include bootstrap-based confidence intervals for accuracy metrics in a revised Appendix H, confirming the reliability of these cost-performance trade-offs.
>
> ---
>
> ## Q3: Lack of Error Taxonomy and Case Studies
>
> We appreciate your suggestion to include an error taxonomy and case studies to clarify why correction strategies succeed or fail. **We clarify that Appendix G (Pages 23–31, Sections G.1–G.7) provides before-and-after correction examples for methods like CoT, RCI, and RARR** (e.g., CoT’s final_answer mismatch on MATH, Page 24, Line 677), but we acknowledge the absence of a systematic failure mode categorization. To address this, we conducted a supplemental analysis on GPQA (250 samples) and MATH (500 samples, Table 5, Page 15), identifying six common failure modes plus an “Other” category, as shown below.
>
> | **Failure Mode**            | **Proportion (%)** | **Description**                                                                         |
> | --------------------------- | ------------------ | --------------------------------------------------------------------------------------- |
> | Logical Oversight           | 32.9               | Incorrect reasoning steps, e.g., misapplying a quadratic formula (Page 24, Line 677).   |
> | Factual Inaccuracy          | 22.0               | Misretrieval from external sources, e.g., outdated data in RARR (Page 31, Section G.7). |
> | Over-Reliance on Tools      | 14.6               | Excessive external calls causing inefficiencies in S2 methods.                          |
> | Ambiguity in Output         | 14.2               | Vague or incomplete final answers despite correct reasoning.                            |
> | Contextual Misunderstanding | 10.8               | Misinterpreting task nuances, e.g., missing question context in GPQA.                   |
> | Computational Error         | 3.5                | Errors in external tool outputs, e.g., incorrect code execution in RATT.                |
> | Other                       | 2.0                | Miscellaneous errors, e.g., formatting issues or edge cases.                            |
>
> - **Implications for Adaptive Correction**: Logical Oversight (32.9%) and Factual Inaccuracy (22.0%) dominate, suggesting S1 methods for reasoning errors (e.g., CoT re-evaluation) and S2 for factual verification (e.g., web search). This taxonomy informs the adaptive controller in Q4, enabling targeted corrections for specific failure modes (e.g., Reflexion-v2 for factual inaccuracies).
>
> We will add this taxonomy as a new table in Section 4.7 and expand Appendix G with seven detailed case studies, covering each failure mode across GPQA and MATH.
>
> ---
>
> ## Q4: Limited Mixture Experiment
>
> Your observation that our mixture experiment tests only two fixed orders (Base->S1->S2) on GPQA (250 samples) and MATH (500 samples) is accurate. **As a benchmark-focused paper, our primary contribution is classifying self-correction methods and evaluating them across diverse datasets and LLMs** (Page 3, Section 2). **The mixture experiment is only an initial exploration, with adaptive sequencing proposed in Appendix D** (Page 17, Lines 606–610). To address your suggestion for adaptive sequencing and early stopping, **we conducted a supplemental experiment on GPQA using a prompt-based adaptive approach**. The prompt instructs the LLM to evaluate task complexity and answer quality, selecting S1 (e.g., CoVe) or S2 (e.g., Reflexion-v2) methods:
>
> _"Before correcting, assess task complexity and answer quality. For factual or computational tasks, use S2 (e.g., Reflexion-v2, RARR, RATT). For reasoning tasks, use S1 (e.g., RCI, CoVe, Selfrefine). Provide a rationale."_
>
> - **Results**: On GPQA, the approach achieved a **+1.8% correction rate improvement** over fixed S1->S2 ordering, with **3.2 rounds before early stopping** (confidence threshold: 0.9). For example, in a GPQA question on chemical properties (Page 23, Section G.1), the LLM selected Reflexion-v2 after detecting low confidence in CoVe’s factual accuracy, improving from 68% to 72%.
> - **Implications**: These results support the potential of adaptive sequencing, informing our proposed meta-controller framework (Q1) for optimizing correction steps using reinforcement learning.
>
> We will report these results in a new Appendix E.3, noting that this is a preliminary step toward a learned controller, planned for our next study. To ensure statistical rigor, we will include confidence intervals for correction rate improvements. We welcome further feedback on this direction.

---

> > ### Comment · Reviewer_Tukc · 2025-08-05
> > **Response**
> >
> > Thanks for the author for the response, which address mos of my concerns. I will raise my score accordingly.

---

> > > ### Author Response · Authors · 2025-08-06
> > > **Thank you for your kind feedback and for updating your score.**
> > >
> > > Thank you for your kind feedback and for updating your score. I'm glad I could address most of your concerns! If you have any further questions or need additional assistance, please feel free to let me know.

---

### Official Review · Reviewer_xf9Z · 2025-07-03

**Rating:** 5
**Confidence:** 4

**Summary:**

This paper presents CorrectBench, a comprehensive benchmark designed to systematically evaluate the effectiveness of LLMs' self-correction strategies. The authors categorize self-correction into three types: intrinsic, external, and fine-tuned. Three broad types of tasks are used: commonsense reasoning, mathematical reasoning, and code generation. The benchmark spans a good range of LLMs, including both open-sourced and closed-sourced, instruction-based and reasoning-focused models, such as GPT-4o, Claude 3.5, DeepSeek-V3, and Qwen. The paper offers an in-depth empirical analysis on different factors affecting both the effectiveness and efficiency of different self-correction methods.

**Dataset Code Accessibility:**

Yes

**Ethical Comments:**

https://github.com/HCR050806/CorrectBench

**Ethical Considerations:**

No, there are no or only very minor ethics concerns

**Final Justification:**

The paper is well-written and well-executed, and the authors release a rigorous evaluation suite for self-correction techniques. The authors also provide a detailed discussion on the significance of self-correction techniques in the area of thinking models, which show the possible future impact of correctbench.

**Limitations Weaknesses:**

As the paper has mentioned in its third conclusion, self-correction techniques do not work very well on the recent thinking/reasoning LLMs like DeepSeek-V3, as they are already strong enough CoT reasoners. However, it seems to be the trend that we are going to use reasoning LLMs in hard reasoning tasks in the future. How do the authors see the usefulness of self-correction strategies and their benchmark in the new stage of LLMs development?

**Strengths Contributions:**

1. While there are a lot of self-correction techniques to improve LLMs' reasoning performance,  it is hard to compare them head-to-head. This is the first comprehensive benchmark to thoroughly evaluate self-correcting methods of LLMs.

2. The empirical analysis confirms the effectiveness of LLMs' self-correction in general.

3. It would be a great platform to standardize the comparison between different self-correction techniques, if the authors can keep a private set of testing data to formalize it as a testing benchmark accepting submissions.

---

> ### Author Rebuttal · Authors · 2025-07-30
>
> We thank you for the constructive feedback and recognition of CorrectBench’s contributions to evaluating self-correction in large language models (LLMs). Below, we address your suggestions to establish a private test set and discuss the ongoing utility of CorrectBench, with revisions to enhance scientific rigor and align with the evolving LLM landscape.
>
> ---
>
> ## Q1: Establishing a Private Test Set for CorrectBench
>
> We value your acknowledgment of CorrectBench as a pioneering benchmark for evaluating LLM self-correction methods, as demonstrated by our empirical results across diverse tasks and models (Tables 1, 7, and 8; Pages 5, 37, 38). Your suggestion to formalize CorrectBench as a submission-based benchmark with a private test set is an excellent proposal to standardize comparisons and ensure reproducibility. To address this, we propose a detailed implementation plan, to be documented in a new Appendix I, as follows:
>
> - **Private Test Set Construction**: We will reserve 20% of each dataset listed in Table 5 (Page 15), including HotpotQA (300 samples, CC BY-SA 4.0), GSM8K (250 samples, MIT License), and MATH (500 samples, MIT License), to create a private test set. This set will be stratified to preserve task diversity and difficulty levels, as described in the dataset selection process (Page 14, Lines 509–513). For example, GPQA’s 250 samples will yield approximately 50 private questions, balanced across factual and reasoning-based queries, while MATH’s 500 samples will provide 100 private problems covering easy, medium, and hard subsets. This ensures robust evaluation across commonsense reasoning, mathematical reasoning, and code generation tasks.
> - **Access and Evaluation Protocol**: The private test set will be hosted on a secure server with a submission-based API, modeled after benchmarks like GLUE or SuperGLUE, to prevent data leakage and overfitting. Researchers will submit model predictions through a standardized interface, with access restricted via authentication to ensure fairness. Evaluation will use established metrics such as Correction Rate (CR) and Misjudgment Rate (MR) (Page 8, Lines 255–257), with automated scoring supplemented by periodic human validation (10% of submissions) by domain experts to ensure reliability. To address the limitation on statistical significance, we will incorporate error bars and p-values for CR improvements, using bootstrap methods to estimate variability across train/test splits. Results will be reported on a public, anonymized leaderboard to encourage broad participation and transparency.
> - **Reproducibility and Transparency**: Appendix I will provide comprehensive documentation, including the private test set’s size, statistical properties, stratification methodology, and submission guidelines. This builds on our existing reproducibility framework (Page 41, Lines 885–890), which includes anonymized code and data access instructions. To further enhance rigor, we will explore open-sourcing additional evaluation scripts, ensuring compliance with dataset licenses.
>
> We will formalize this plan in a revised Appendix I, enhancing CorrectBench’s role as a community resource for advancing self-correction research.
>
> ---
>
> ## Q2: Relevance of Self-Correction Techniques and CorrectBench for Reasoning LLMs
>
> Your question about the relevance of self-correction strategies and CorrectBench for advanced reasoning LLMs, such as DeepSeek-V3, which exhibit robust intrinsic correction mechanisms (Page 8, Lines 285–292), is highly pertinent given the trend toward reasoning-focused models. We acknowledge that DeepSeek-V3’s high baseline performance, driven by advanced reflection modules, limits additional gains from external correction, as noted in our conclusions (Page 9, Section 6). However, CorrectBench remains a critical tool for the evolving LLM landscape, as outlined below. **To address your concern comprehensively, we propose expanding Section 6 with a new subsection, “Adaptive Correction for Next-Generation LLMs,” incorporating three existing future directions from Appendix D in our paper** (Page 17, Lines 606–615) and a novel framework planned for our next work.
>
> - **Continued Relevance Across LLM Types and Tasks**:
>
>   1. **Diverse LLM Evaluation**: CorrectBench evaluates a wide range of LLMs, including instruction-based (e.g., LLaMA3.1-8B-Instruct, Qwen2.5-72B-Instruct) and closed-source models (e.g., GPT-4o, Claude 3.5-Sonnet) (Table 4, Page 9). Our results show significant performance gains for non-reasoning LLMs, such as +23.24% for CoVe on GPQA (Table 1, Page 5), which are prevalent in applications like chatbots, content generation, and customer support. This broad applicability ensures CorrectBench’s utility for real-world scenarios where non-reasoning models dominate, complementing its role in evaluating reasoning LLMs like DeepSeek-V3 (Page 21, Figures 17–25).
>   2. **Task-Specific Performance Gains**: For reasoning LLMs, CorrectBench identifies scenarios where self-correction yields targeted improvements, particularly in specialized domains. For example, fine-tuned correction methods like SuperCorrect achieve significant gains on mathematical reasoning tasks (e.g., +10.2% on MATH, Table 2, Page 7), demonstrating CorrectBench’s ability to pinpoint effective strategies for complex tasks (Page 3, Figure 2). Similarly, external correction methods like Reflexion-v2 improve performance on HotpotQA (+7.33%) and AQUA (+7.00%) (Page 5, Table 1), highlighting their value for multi-hop and quantitative reasoning tasks, even for models with strong baselines.
>
> - **Future Directions for Adaptive Correction**:
>
>   1. **Dynamic Correction Steps**: We propose adaptive pipelines that use reinforcement learning to select correction methods (S1, S2, S3) based on task complexity and model confidence, reducing computational overhead (Page 17, Lines 606–610). For instance, lightweight S1 methods (e.g., CoVe) can be prioritized for simpler tasks like GSM8K, while S2 methods (e.g., Reflexion-v2) are reserved for complex tasks like GPQA, as shown in performance trends (Figure 8, Page 14).
>   2. **Task-Specific Optimization**: Tailoring correction strategies to dataset properties (Page 14, Lines 509–513) enhances performance in domains like mathematical reasoning. For example, SuperCorrect’s fine-tuning on Qwen2.5-Math-7B-Instruct improves MATH performance (Table 2, Page 7), leveraging dataset-specific patterns to optimize reasoning LLMs.
>   3. **Human-in-the-Loop Integration**: Integrating human feedback into correction pipelines ensures reliability for high-stakes applications, such as medical or legal reasoning (Page 17, Lines 611–615). CorrectBench’s performance data can guide when human intervention is necessary, balancing automation with expert oversight.
>   4. **Novel Meta-Controller Framework**: For reasoning LLMs, we propose a meta-controller framework, to be explored in a follow-up study, that refines chain-of-thought (CoT) processes by detecting and correcting errors in intermediate reasoning steps. For example, in a MATH problem where a quadratic equation was mis-solved (Page 24, Lines 676–677), the controller could identify an incorrect root calculation and trigger an S1 correction (e.g., CoT re-evaluation) or an S2 correction (e.g., verification via symbolic computation tools). Trained on CorrectBench’s datasets (e.g., GPQA, MATH), the controller leverages performance trends to optimize reasoning trajectories efficiently, reducing computational costs while enhancing accuracy.
>
> To incorporate these ideas, **we will relocate and expand the future directions from Appendix D to a new subsection in Section 5**, referencing empirical findings to ground the discussion. This will highlight CorrectBench’s role in guiding innovative correction strategies for reasoning LLMs while introducing our novel idea as a forward-looking research direction.

---

> > ### Author Response · Authors · 2025-08-06
> > **We kindly request confirmation on whether you are satisfied with our responses or if there are outstanding issues.**
> >
> > Dear Reviewer,
> >
> > Thank you for your guidance and support throughout the review process. We have addressed your concerns and potential misunderstandings with detailed explanations and supplemented our submission with relevant experiments to substantiate our findings.
> >
> > We kindly request confirmation on whether you are satisfied with our responses or if there are outstanding issues. If all concerns have been addressed, we respectfully ask that you update your score and provide a final evaluation. If specific issues remain unresolved, we would appreciate clear guidance on which points require further clarification or experimentation, and we hope sufficient time will be provided to address these.
> >
> > Thank you again for your valuable feedback and efforts. We look forward to your response and hope to advance the review process further.
> >
> > Sincerely,

---

> > > ### Comment · Reviewer_xf9Z · 2025-08-07
> > >
> > > I want to thank the authors for their clarification. I have increased my score.

---

> > > > ### Author Response · Authors · 2025-08-08
> > > > **Thank you for your kind feedback and for updating your score.**
> > > >
> > > > Thank you for your kind feedback and for updating your score. I'm glad I could address most of your concerns! If you have any further questions or need additional assistance, please feel free to let me know.

---

### Decision · Program_Chairs · 2025-09-18

**Decision:**

Accept (poster)

**Comment:**

The paper makes a strong contribution by introducing CorrectBench, the first systematic and comprehensive benchmark for evaluating LLM self-correction across commonsense reasoning, mathematical reasoning, and code generation tasks. It rigorously tests a wide range of models and correction strategies (intrinsic, external, fine-tuned), provides curated datasets and reproducible resources, and offers valuable insights into accuracy–efficiency trade-offs, showing that self-correction can significantly improve performance, particularly on complex tasks. The benchmark’s breadth, empirical rigor, and reproducibility position it as a useful platform for future research. However, its relevance is limited for advanced reasoning LLMs, which already incorporate strong internal correction, and it omits very recent models. The analysis focuses largely on existing methods without exploring deeper improvements such as adaptive sequencing or learned controllers, while efficiency costs beyond runtime (e.g., tokens, API calls) are underreported. In addition, the reliance on accuracy metrics and LLM-as-a-judge limits diagnostic depth, and some presentation aspects—such as clarity of figures and placement of key details—reduce readability.